# Use of Genetic Programming for the Estimation of CODLAG Propulsion System Parameters

**Nikola Anđelić** [1], **Sandi Baressi Šegota** [1], **Ivan Lorencin** [1], **Igor Poljak** [2], **Vedran Mrzljak** [1,*] **and Zlatan Car** [1]

1 Faculty of Engineering, University of Rijeka, Vukovarska 58, 51000 Rijeka, Croatia; nandelic@riteh.hr (N.A.); sbaressisegota@riteh.hr (S.B.Š.); ilorencin@riteh.hr (I.L.); car@riteh.hr (Z.C.)
2 Maritime Department, University of Zadar, Mihovila Pavlinovića 1, 23000 Zadar, Croatia; ipoljak1@unizd.hr
* Correspondence: vmrzljak@riteh.hr; Tel.: +385-98-174-5205

**Abstract:** In this paper, the publicly available dataset for the Combined Diesel-Electric and Gas (CODLAG) propulsion system was used to obtain symbolic expressions for estimation of fuel flow, ship speed, starboard propeller torque, port propeller torque, and total propeller torque using genetic programming (GP) algorithm. The dataset consists of 11,934 samples that were divided into training and testing portions in an 80:20 ratio. The training portion of the dataset which consisted of 9548 samples was used to train the GP algorithm to obtain symbolic expressions for estimation of fuel flow, ship speed, starboard propeller, port propeller, and total propeller torque, respectively. After the symbolic expressions were obtained the testing portion of the dataset which consisted of 2386 samples was used to measure estimation performance in terms of coefficient of correlation ($R^2$) and Mean Absolute Error ($MAE$) metric, respectively. Based on the estimation performance in each case three best symbolic expressions were selected with and without decay state coefficients. From the conducted investigation, the highest $R^2$ and lowest $MAE$ values were achieved with symbolic expressions for the estimation of fuel flow, ship speed, starboard propeller torque, port propeller torque, and total propeller torque without decay state coefficients while symbolic expressions with decay state coefficients have slightly lower estimation performance.

**Keywords:** CODLAG; data-driven modelling; genetic programming; decay state coefficients

## 1. Introduction

The marine propulsion systems are used to generate thrust to propel a ship across the water, with various types of marine prime movers being used [1,2]. The gas turbines are often used in combination with other types of propulsion systems due to their poor thermal efficiency at low power output. The other key factor for using such propulsion systems is to allow a reduction of emissions in sensitive environmental areas or while in port [3]. In some cases, ships have steam turbines which are also used to improve the efficiency of gas turbines in a combined cycle, where waste heat from gas turbine exhaust is used to boil water and create steam.

The combined diesel-electric and gas (CODLAG) is a modified diesel and gas propulsion system for ships. In it, the electric motors which are powered by diesel generators are connected to the propeller shafts. To achieve higher speed, the gas turbine is used to power shafts over a cross-connecting gearbox. For cruise speed, the drive train of the turbine is disengaged with clutches. Since electric motors work efficiently over a wide range of revolutions they can be directly connected to the propeller shaft so simpler gearboxes are used for combining the mechanical output of the turbine and diesel-electric system.

*Literature Review*

The most commonly used maintenance approach was to repair systems as necessary [4]. This approach in the long run proved to be very expensive especially when

gathering data from the field is cheaper and breakdown-related costs may overcome the asset value [5]. Condition-based maintenance (CBM) is triggering maintenance activities as they are indicated by the condition of the system [4]. This approach tracks the condition of system parts which is used to predict their potential degradation and to plan when maintenance activities will be performed. To perform accurate fault prognosis the CBM requires real-time tracking and diagnosis of the target system.

The comprehensive approach in the simulation of CODLAG propulsion system behavior during transients and off-design conditions is presented by Altosole et al. (2010) [6]. With this model, the authors were able to capture the unbalance of the shaft line during a turning maneuver. The influence of the deterioration of the main components (gas turbine, propellers, and ship hull) on the behavior of the CODLAG propulsion system was performed in [7]. The different detailed simulation models of the CODLAG propulsion system were developed by Martelli (2017) [8] to investigate the system performance under different operational conditions. The publicly available dataset has been developed using numerical simulation of CODLAG propulsion plant [9], where the performance advantages of exploiting machine learning (ML) methods in modeling the degradation of the propulsion plant over time are tested. In [10], the multi-layer perceptron (MLP) was applied on data available dataset in the prediction of the gas turbine and turbo compressor decay state coefficients. In the case of gas turbine decay state coefficient prediction, the lowest mean relative error of 0.622% was achieved while in the case of turbo compressor decay state coefficient, the lowest mean relative error of 1.094% was achieved. In [11], the MLP was again used for the estimation of the frigate speed. The results showed that MLP could estimate the shipping speed with an error of just $3.4485 \times 10^{-5}$ knots. In [12], the publicly available CODLAG dataset was used to train genetic programming algorithm to obtain symbolic expressions for estimation gas turbine shaft torque and fuel flow. The three best symbolic expressions obtained for gas turbine shaft torque estimation generated $R^2$ scores of 0.999201, 0.999296, and 0.999374, respectively. The three best symbolic expressions obtained for fuel flow estimation generated $R^2$ scores of 0.995495, 0.996465, and 0.996487, respectively.

Beyond the aforementioned papers, many researchers opted for an application of AI-based modeling techniques in the application in propulsion system research area. Cheliotis et al. (2020) [13] demonstrate the application of Exponentially Weighted Moving Average (EWMA) for fault detection in maritime systems. The proposed research achieves an $R^2$ score of 0.96 in both observed cases. Uyanik et al. (2020) [14] proposed an ML approach to the prediction of a container vessel fuel consumption. Through the application of multiple algorithms, such as Multiple Linear Regression, Ridge and LASSO Regression, Support Vector Regression, Tree-Based Algorithms, and Boosting Algorithms are applied and evaluated using $R^2$. The best results are achieved through multiple linear regression and ridge regression with an $R^2$ value of 0.999. Berghout et al. (2021) [15] applied an Extreme Learning Machine in combination with other techniques in the application for prediction of condition-based maintenance of naval propulsion systems. The newly proposed approach demonstrates not only higher accuracy, but also better generalization under different training paradigms. Tsaganos et al. (2020) [16] demonstrated the application of AdaBoost classifier for the improvement of engine fault detection. Based on the achieved performance, with an accuracy of 96.5%, the authors concluded that the ensemble methods such as used are an appropriate choice for the given problem. Bachmayer et al. (2020) [17] discussed ML applications in underwater propulsion systems, concluding that such approaches are fast enough for use in the real-time system for detection of soft and hard errors.

GP is an Artificial Intelligence (AI) method for evolving expressions such as computer programs or equations. The roots of GP can be traced back to Alan Turing [18] but the computational limitations of that time prevented further development. After almost 30 years the small programs were successfully evolved, as reported in [19]. The genetic algorithm (GA) for evolving programs was officially introduced by Koza in 1988 [20]. The

algorithm can be used to develop symbolic ecpressions which allow for direct modelling of various tasks [21–23].

Based on an extensive literature review the following questions arise:

- does the correlation exist, and how strong is the correlation between the parameters of CODLAG propulsion system dataset [9], and
- is it possible to obtain the symbolic expressions using GP algorithm for fuel flow estimation, ship speed estimation, starboard and port propeller torque, and total torque-with and without decay state coefficients.

The correlation analysis will give a better insight into the CODLAG propulsion system dataset [9] which will be a good starting point for GP algorithm implementation. After the symbolic expressions were obtained and tested the results of correlation analysis will provide sufficient information in further investigation of symbolic expressions.

The novelty of the research lies in multiple elements. The authors have applied the correlation analysis to determine the parameter importance of individual dataset parameters, in order to improve the results of the AI-based methods. The main novelty of the paper is the generation of equations which can be applied to the prediction of the aforementioned parameters (fuel flow, ship speed, as well as starboard, port and total propeller torque) by the future researchers. As a final research novelty, the influence of decay coefficients has been tested.

First, the researchers will present the used dataset, with methods applied to the analysis of it. Then, a short description of the GP algorithm is provided, along with the used hyperparameters and evaluation metrics. The results are presented and discussed; following that, providing information on the correlation coefficients of the parameters in the dataset, metrics achieved with the trained models along with the used hyperparameters and regressed equations. Drawn conclusions, addressing the posed research questions, are given in the end.

## 2. Materials and Methods

In this section, the publicly available dataset [9] is described in detail as well as the correlation analysis, genetic programming algorithm, and metric used to evaluate obtained symbolic expressions.

### 2.1. Dataset Description

The dataset that was used in this paper is a publicly available dataset available at the UCI machine learning repository [9]. The dataset was obtained using a numerical simulator of a naval vessel (Frigate) characterized by a Gas Turbine (GT) propulsion plant. The simulator that was used to obtain the dataset consists of different blocks such as propeller, hull, GT, gearbox, and controller. These components were developed and fine-tuned on several similar real propulsion plants. This dataset also incorporates the performance decay over time of the GT components such as turbo compressors and turbines. The two propellers are driven from power generated with GT and two electric motors which are transmitted using a system that consists of three gearboxes and four clutches. The scheme of the CODLAG propulsion system is shown in Figure 1.

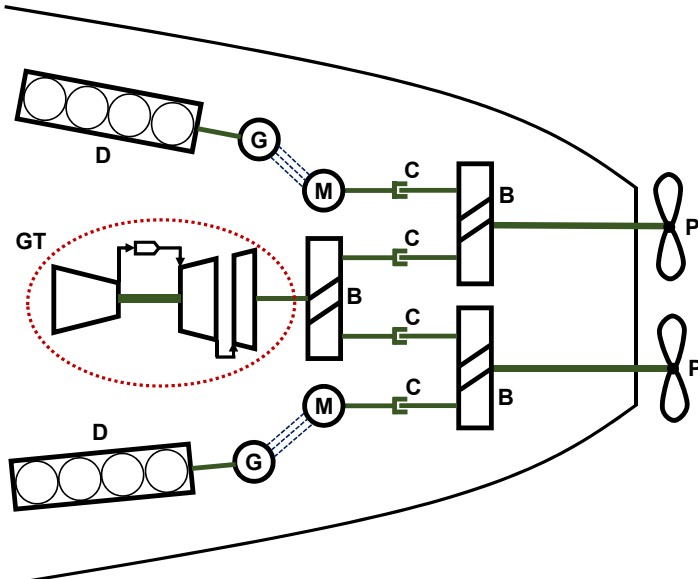

**Figure 1.** The scheme of CODLAG propulsion system (B-gear box, C-clutch, D-diesel engine, G-electrical generator, GT- gas turbine, M-electrical motor, P-frigate propeller).

The GT shown in Figure 1 consists of a turbo compressor, combustion chamber, high pressure (HP), and low pressure (LP) gas turbines. It should be noted that the power produced in HP gas turbine is used only for turbo compressor drives (gas generator) while the power produced by LP gas turbine is used for ship propulsion in combination with power produced by electric motors. The detailed scheme of GT used in the CODLAG propulsion system is shown in Figure 2.

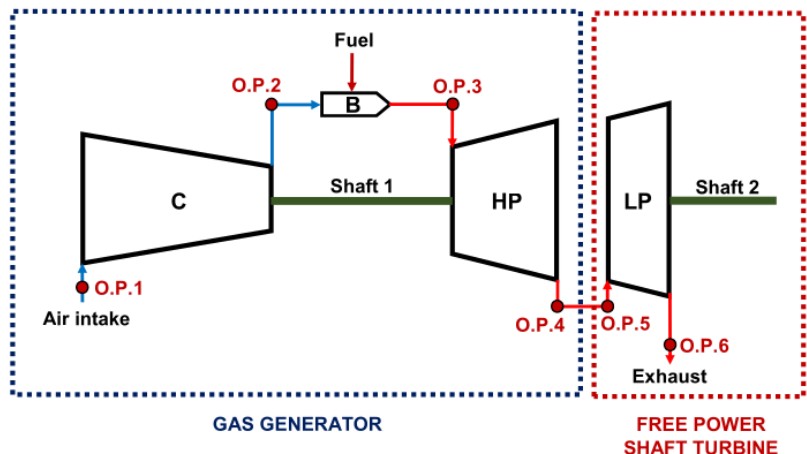

**Figure 2.** The scheme of GT component used in CODLAG propulsion system (C-turbo compressor; B-combustion chamber; HP-high pressure turbine; LP-low pressure turbine, O.P.-Operating Point).

As seen in Figure 2, the HP gas turbine together with turbo compressor (C) and combustion chamber (B) represents the gas generator. The only connection between HP gas turbine and LP gas turbine is achieved by flue gases that go from HP gas turbine to LP gas turbine. The LP gas turbine is a free power shaft turbine. System maintenance is an important factor of complex propulsion systems. To describe the gas turbine and turbo compressor the decay state coefficient is used as the numerical indicator of their condition. In this dataset, the decay state coefficients of gas turbine and turbo compressor are simulated in the MatLab software package as the consequence of fouling. The source of fouling is the exhaust gases and oil vapors that produce impurities on gas turbine blades

and impurities of intake air of turbo compressor. The fouling in the gas turbine is simulated as the gas flow rate decrease while in the turbo compressor the fouling is simulated as a decrease of airflow rate $M_c$ and isentropic efficiency $\eta_c$. In Table 1 the dataset parameters with corresponding values range and units are provided, while Figure 3 shows the T-s diagram of the Gas turbine for the CODLAG system.

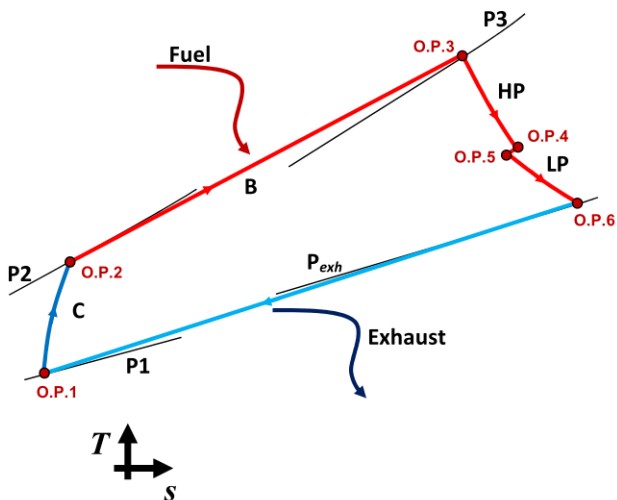

**Figure 3.** Thermodynamic process of the gas turbine from the analyzed CODLAG propulsion system in T-s diagram (O.P.-Operating Point).

**Table 1.** The list of physical values in CODLAG dataset with corresponding range of values and units.

| Physical Variable | Range | Unit |
| --- | --- | --- |
| Lever position ($l_p$) | 1.138–9.3 | - |
| Ship speed ($v$) | 3–27 | kn |
| Gas turbine shaft torque (GTT) | 253.547–72,784.872 | kNm |
| GT rate of revolutions (GTn) | 1307.675–3560.741 | rpm |
| Gas generator rate of revolutions (GGn) | 6589.002–9797.103 | rpm |
| Starboard propeller torque (Ts) | 5.304–645.249 | kN |
| Port propeller torque (Tp) | 5.304–645.249 | kN |
| High pressure turbine exit temperature (T48) | 442.364–1115.797 | °C |
| Turbo compressor inlet air temperature (T1) | 288 | °C |
| Turbo compressor outlet air temperature (T2) | 540.442–789.094 | °C |
| HP turbine exit pressure (P48) | 1.093–4.56 | bar |
| Turbo compressor inlet air pressure (P1) | 0.998 | bar |
| Turbo compressor outlet air pressure (P2) | 5.828–23.14 | bar |
| GT exhaust gas pressure ($P_{exh}$) | 1.019–1.052 | bar |
| Turbine injection control (TIC) | 0–92.556 | % |
| Fuel flow ($m_f$) | 0.068–1.832 | kg/s |
| Turbo compressor decay state coefficient | 0.95–1 | - |
| Turbine decay state coefficient | 0.975–1 | - |

### 2.2. Correlation Analysis

In this paper, two types of correlation analysis will be applied to the CODLAG propulsion system dataset to determine the correlation between input and output variables i.e., Pearsons and Spearman correlation analysis.

The Pearson's product-moment correlation coefficient $r$ measures the linear relationship between two continuous variables [24]. For example, let $x$ and $y$ represent the

quantitative measures of two random variables on the same sample of $n$. The Pearson's correlation coefficient $r$ can be written in the following form [25]:

$$r = \frac{\sum_{i=1}^{n}(x_i - \bar{x})(y_i - \bar{y})}{\sqrt{\sum_{i=1}^{n}(x_i - \bar{x})}\sqrt{\sum_{i=1}^{n}(y_i - \bar{y})}} \tag{1}$$

where

$$\bar{x} = \frac{1}{n}\sum_{i=1}^{n}x_i \quad \text{and} \quad \bar{y} = \frac{1}{n}\sum_{j=1}^{n}y_i \tag{2}$$

are the mean values of variable $x$ and $y$, respectively. Assuming that the sample variances of $x$ and $y$ are positive i.e., $s_x^2 > 0$ and $s_y^2 > 0$ the linear correlation coefficient $r$ can be written as the ratio of the sample covariance of the two variables to the product of their respective standard deviations $s_x$ and $s_y$ as [26,27]:

$$r = \frac{\text{Cov}(x,y)}{s_x s_y}, \tag{3}$$

where Cov represents covariance. The range of correlation measurement $r$ is between $-1$ and $+1$. There are three different cases of correlation measurement between $x$ and $y$ and these are:

- $r > 0$-the linear correlation between $x$ and $y$ are positive i.e., higher absolute levels of one variable are associated with lower levels of the other,
- $r = 0$-indicates the absence of any association between $x$ and $y$, and
- $r < 0$-the linear correlation between $x$ and $y$ is negative i.e., higher absolute levels of one variable are associated with lower levels of the other.

The magnitude of the correlation coefficient indicates the strength of association, while the sign of the linear correlation coefficient indicates the direction of the association. For example, if the value of the correlation coefficient is equal to +1 the variables have a perfect linear positive correlation which means that if one variable increases, the second increases proportionally in the same direction. On the other hand, if the correlation coefficient value is equal to $-1$, the variables have a negative correlation and move in the opposite direction of each other. If the value of one variable increases the value of the other variable decreases proportionally. When two variables $x$ and $y$ are normally distributed, the population Pearson's product-moment correlation coefficient can be determined as [28]:

$$\rho = \frac{\text{Cov}(x,y)}{\sigma_x \sigma_y}, \tag{4}$$

where $\sigma_x$ and $\sigma_y$ are the population standard deviations of $x$ and $y$, respectively. It should be noted that if both variables are normally distributed the coefficient $\rho$ is not significant since it is affected by extreme values.

Spearman's correlation coefficient evaluates the monotonic relationship between two continuous variables [29]. In a monotonic relationship, the variables tend to change together, but not at constant rate. For two variables $x$ and $y$ the Spearman's rank correlation coefficient computes the correlation between the rank of two variables which can be written in the following form [30]:

$$r_s = \frac{\sum_{i=1}^{n}(x_i' - \bar{x}')(y_i' - \bar{y}')}{\sqrt{\sum_{i=1}^{n}(x_i' - \bar{x}')}\sqrt{\sum_{i=1}^{n}(y_i' - \bar{y}')^2}} \tag{5}$$

where $x'$ and $y'$ are ranks of $x$ and $y$, respectively. The Spearman's correlation is basically the rank-based version of the Pearson's correlation coefficient. The range of Spearman's coefficient is from $-1$ up to $+1$. Similar to Pearson correlation coefficient, the Spearman's correlation coefficient is 0 for variables that are correlated in a non-monotonic way. An

alternative formula used to calculate the Spearman rank correlation can be written in the following form [31]:

$$r_s = 1 - \frac{6\sum_{i=1}^{2} d_i}{n(n^2 - 1)},$$

(6)

where $d_i$ is the difference between the ranks of corresponding values $x_i$ and $y_i$. To avoid the step of determining the ranks of the variables, Equation (5) was used for the calculation of Spearman's correlation coefficients in this paper.

### 2.3. Genetic Programming

The genetic programming algorithm is a technique of evolving programs from an initial population of random, unfit programs from generation to generation and fits them for a particular task with the application of genetic operations (crossover and mutation) [32]. In GP computer programs are represented as three structures. The example of computer program $(X_1 + 2.7X_2) + (X_3 - 3.7X_4)$ is shown in Figure 4.

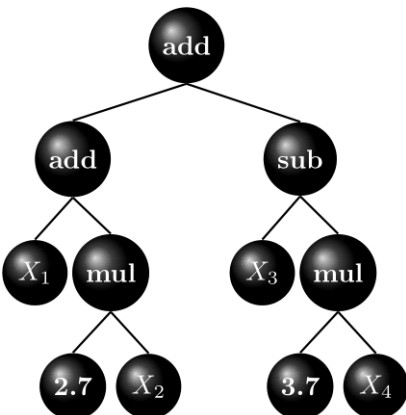

**Figure 4.** The example of computer program represented as three structure.

The variables and constants shown in Figure 4 are leaves of the tree and in GP they are called terminals while the arithmetic operations are internal nodes called functions. The set of functions and terminals together form the primitive set of a GP system.

As stated earlier the initial population consists of random, naive programs which are developed using a primitive set. Various methods can be used to initialize the population however in this paper the ramped-half-and-half method is used. This method is a combination of the full and grow method. In the full method, the nodes are taken at random from the function set until the maximum tree depth is reached. After the maximum tree depth is reached only terminals must be chosen. In grow method the nodes are selected from the whole primitive set until the depth limit is reached. Once the depth limit is reached only terminals may be chosen. Since both methods do not provide a very wide array of sizes and shape the ramped half-and-half method is used. In this method, half of the initial population is generated using the full method, and the other half using the grow method. This procedure is done using a range of depth limits to ensure that the variety of tree sizes and shapes in population. After the initial population is generated each population member must be evaluated to determine its fitness value. In this paper, the Mean Absolute Error is a fitness measure that will be used to evaluate each population member. The $MAE$ formula can be written in the following form [33]:

$$MAE = \frac{\sum_{i=1}^{n} |y_i - x_i|}{n},$$

(7)

where $y_i$ is prediction and $x_i$ is the true value thus the difference between those two values represents an average of the absolute errors while $n$ represents the number of samples. It should be noted that this measure will also be used later for further evaluation of symbolic

expressions on the testing portion of the dataset. After the initial population has been created the selection must be performed to select population members that will represent parents of the next generation. There are various types of the selection procedure which can be used; however, in this paper, the tournament selection procedure was used. The tournament selection starts from a random selection of population members from all population members [34]. These population members are compared with each other and the best of them (tournament winner) is chosen to be the parent. For crossover operation two parents are needed so, two selection tournaments are made. However, for mutation operation, only one population member (tournament winner) is required so only one tournament selection is required. In GP the most commonly used form of crossover is the subtree crossover. This operation requires two parents and the crossover point or a node is randomly selected in each parent tree. The subtrees are swapped between those two parents to generate the members of the next generation. In GP there are three types of mutation operations and these are subtree mutation, hoist mutation, and point mutation. In each mutation case, only one tournament winner is needed. The subtree mutation starts by randomly selecting the subtree on the tournament winner and this subtree is replaced by a randomly generated subtree to form an offspring of the next generation. The hoist mutation operation starts by randomly selecting the subtree on the tournament winner. Then a random subtree of that subtree is selected and is then hoisted into the original subtree location to form the member of the next generation. The point mutation operation starts by selecting random nodes on the tournament winner which will be replaced. The terminals are then replaced by other terminals and functions are replaced by other functions.

To terminate the execution of the GP algorithm the stopping criteria are needed. Two different stopping criteria are usually used in GP and these are the maximum number of generations and the stopping criteria value. The maximum number of generations is the termination criteria that terminates the execution of GP after the maximum number of generations is reached. The stopping criteria value represents the lowest fitness function value which can be achieved by population members in a generation. If the lowest value is achieved the GP algorithm execution is terminated.

The other important parameter in the GP algorithm is the parsimony coefficient [35] which is responsible for penalizing large growth of symbolic expressions without improvement in their fitness value by making them less favorable for tournament selection.

### 2.4. Evaluation Metrics

After all symbolic expressions were obtained with the GP algorithm on the training portion of the dataset these symbolic expressions are then evaluated on the testing portion of the dataset. In this paper, two metrics are used for the evaluation of estimation performance of symbolic expressions and these are the $R^2$ and $MAE$ metric. Since the $MAE$ was already described in the previous section here only the $R^2$ metric will be described.

The $R^2$ metric or the coefficient of determination is the proportion of the variance in the dependent variable that is predictable from the independent variable. The formula for calculating the $R^2$ metric can be written in the following form

$$R^2 = 1 - \frac{S_{RESIDUAL}}{S_{TOTAL}} = 1 - \frac{\sum_{i=0}^{m}(y_i - \hat{y}_i)^2}{\sum_{i=0}^{m}(y_i - \frac{1}{m}\sum_{i=0}^{m} y_i)^2} \tag{8}$$

Two sets of solutions i.e., the real data $y$ and the data obtained by the model $\hat{y}$ are compared by this metric in terms of variance. The result of $R^2$ metric can be in the range from 0 to 1. If the $R^2$ value is equal to 1.0 means that there is no variance between the real data and the data obtained by the model. The $R^2$ value of 0 means none of the variances in the real data are explained in the model data.

## 3. Results and Discussion

In this section, the preparatory steps for implementation of GP are described as well as the results obtained using correlation analysis and symbolic expressions obtained for estimation of the fuel flow, ship speed, starboard, and propeller torque, and total torque, respectively. After extensive research, the obtained results are discussed in detail.

### 3.1. Results

Before presenting the best symbolic expressions for estimation of specific output values the two types of correlation analysis were performed and these are Pearsons and Spearman's correlation analyses. The results of Pearsons and Spearman's correlation analyses are shown in Figures 5 and 6.

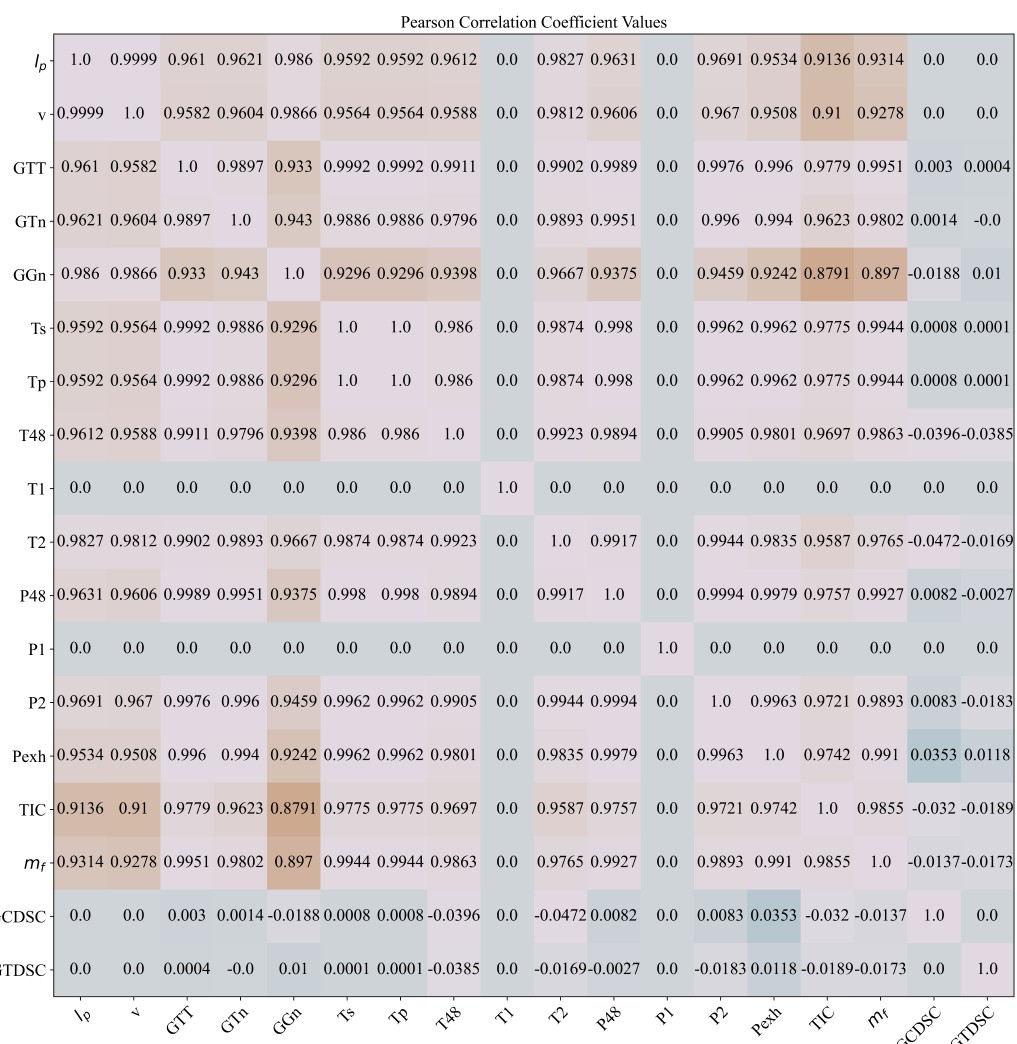

**Figure 5.** The result of Pearsons correlation analysis.

As seen in Figure 5 the highest positive correlation values are obtained for 14 out of 18 variables in the dataset. This means that if the value of these input variables increases the value with the output variable will also increase. However, the GCDSC (turbo compressor decay state coefficient) and GTDSC (turbine decay state coefficient) have positive, negative, and no correlation values with other variables in the dataset. Both decay state coefficients do not correlate with ship speed ($v$), have small positive correlation values (0.0008, 0.0001) with starboard and port propeller torque, and negative correlation values ($-0.0137$, $-0.0173$) with fuel flow. If the correlation value is negative this means that if the value of input variables increases the value of the output variable will decrease or vice versa. It should be

noted that T1 (GT turbo compressor inlet air temperature) and P1 (GT turbo compressor inlet air pressure) have no correlation with any variable in the dataset except with itself. These two variables represent the ambient temperature and pressure which were set to constant values during the simulation of the CODLAG propulsion system. The variation of these two variables would not have any effect on the output variable. The results of Spearman's correlation analysis are shown in Figure 6.

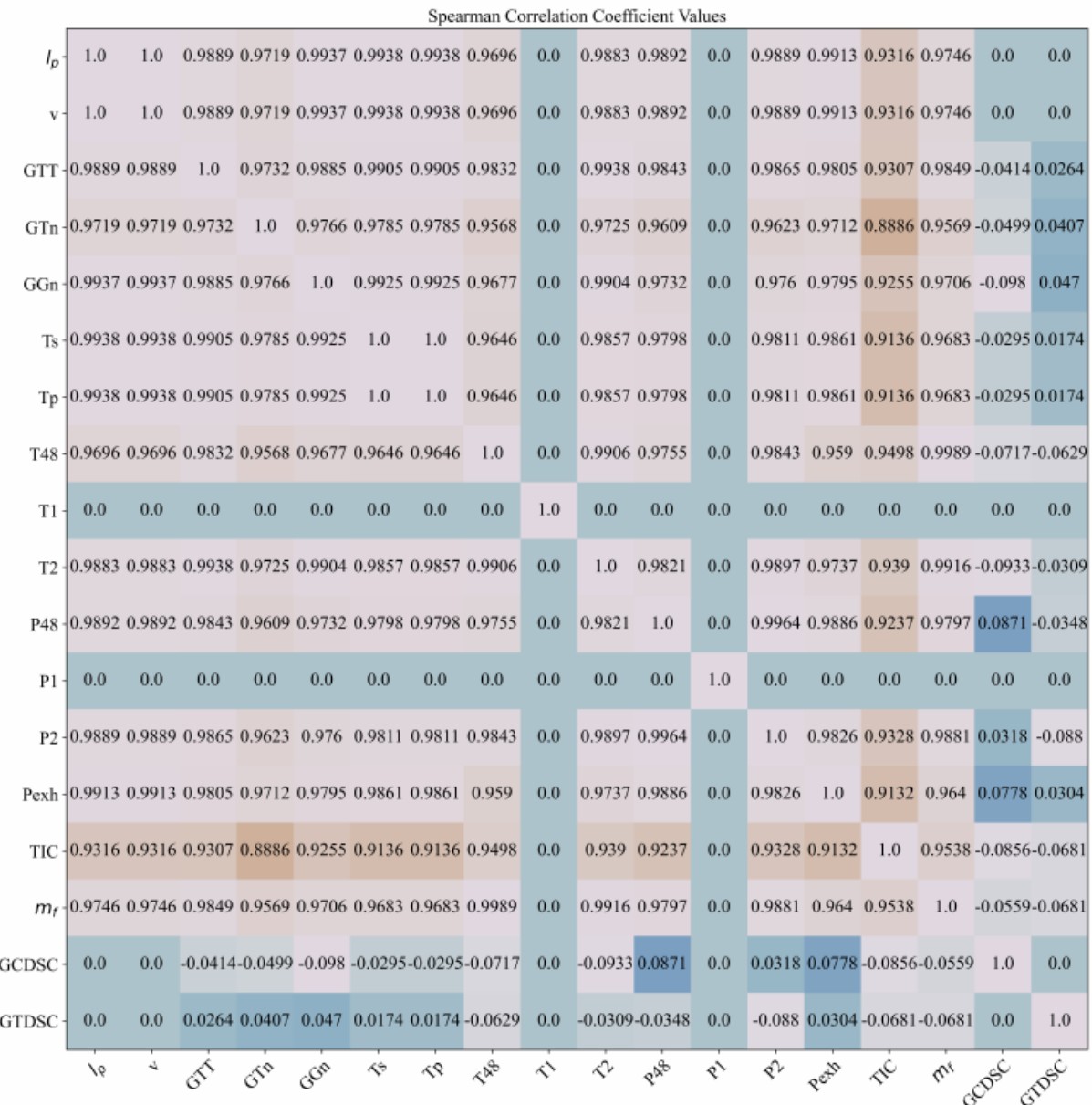

**Figure 6.** The result of Spearman's correlation analysis.

The results of performed Spearman's correlation analyses have similar results as in the case of Pearson's correlation analysis. The correlation analysis showed that 14 out of 18 variables have positive correlation values. The T1 and P1 are constant values throughout the entire dataset so they do not correlate with any other variable except with themselves i.e., the correlation values are zero. The results of correlation analyses also showed that two decay state coefficients (GCDSC and GTDSC) have positive, negative, or no correlation value. As in the case of Pearson's correlation analysis, the two decay state coefficients do not correlate with ship speed (0.0, 0.0), positive and negative correlation values with

starboard ($-0.0295$, $0.0174$) and port propeller torque ($-0.0295$, $0.0174$), and negative correlation values with fuel flow ($-0.0559$, $-0.0681$).

As stated in the abstract and introduction of this paper there is a total of 10 different GP analyses performed and these are fuel flow, ship speed, starboard propeller torque, port propeller torque, and total propeller torque analysis with and without decay state coefficients. It should be noted that for starboard propeller torque analysis the port propeller torque variable will be excluded from the dataset. The same procedure was applied for port propeller torque analysis. For total propeller torque analysis, the starboard and port propeller torque values were added together and excluded from the dataset as input variables. Table 2 shows input and output variables for each of the analyses.

**Table 2.** The input and output variables used in the GP algorithm to obtain symbolic expressions for estimation of fuel flow, ship speed, starboard, port, and total propeller torque with and without decay coefficient.

| Physical Variable | Representation of Variables in GP | | | | |
| --- | --- | --- | --- | --- | --- |
| | Fuel Flow Analysis | Ship Speed Analysis | Starboard Propeller Torque Analysis | Port Propeller Torque Analysis | Total Propeller Torque Analysis |
| Lever position ($l_p$) | $X_0$ | $X_0$ | $X_0$ | $X_0$ | $X_0$ |
| Ship speed ($v$) | $X_1$ | $y$ | $X_1$ | $X_1$ | $X_1$ |
| Gas turbine shaft torque (GTT) | $X_2$ | $X_1$ | $X_2$ | $X_2$ | $X_2$ |
| GT rate of revolutions (GTn) | $X_3$ | $X_2$ | $X_3$ | $X_3$ | $X_3$ |
| Gas generator rate of revolutions (GGn) | $X_4$ | $X_3$ | $X_4$ | $X_4$ | $X_4$ |
| Starboard propeller torque (Ts) | $X_5$ | $X_4$ | $y$ | - | - |
| Port propeller torque (Tp) | $X_6$ | $X_5$ | - | $y$ | - |
| High pressure turbine exit temperature (T48) | $X_7$ | $X_6$ | $X_5$ | $X_5$ | $X_5$ |
| turbo compressor inlet air temperature (T1) | $X_8$ | $X_7$ | $X_6$ | $X_6$ | $X_6$ |
| turbo compressor outlet air pressure (P2) | $X_9$ | $X_8$ | $X_7$ | $X_7$ | $X_7$ |
| HP turbine exit pressure (P48) | $X_{10}$ | $X_9$ | $X_8$ | $X_8$ | $X_8$ |
| Turbo compressor inlet air pressure (P1) | $X_{11}$ | $X_{10}$ | $X_9$ | $X_9$ | $X_9$ |
| Turbo compressor outlet air pressure (P2) | $X_{12}$ | $X_{11}$ | $X_{10}$ | $X_{10}$ | $X_{10}$ |
| GT exhaust gas pressure ($P_{exh}$) | $X_{13}$ | $X_{12}$ | $X_{11}$ | $X_{11}$ | $X_{11}$ |
| Turbine injection control (TIC) | $X_{14}$ | $X_{13}$ | $X_{12}$ | $X_{12}$ | $X_{12}$ |
| Fuel flow ($m_f$) | $y$ | $X_{14}$ | $X_{13}$ | $X_{13}$ | $X_{13}$ |
| Turbo compressor decay state coefficient | $X_{15}$ | $X_{15}$ | $X_{14}$ | $X_{14}$ | $X_{14}$ |
| Trubine decay state coefficient | $X_{16}$ | $X_{16}$ | $X_{15}$ | $X_{15}$ | $X_{15}$ |
| Total Propeller Torque (Ts+Tp) | - | - | - | - | $y$ |

As seen in Table 2 for fuel flow and ship speed analysis there is a total of 16 input variables and one output variable. In the case of fuel flow and ship speed without decay coefficients, there is a total of 14 input variables. In the case of starboard and port propeller torque analysis, there is a total of 13 input variables in the case without decay state coefficient while in the case with decay state coefficients there is a total of 15 input variables. The same number of input variables with and without decay state coefficients is applied for the total propeller torque but the output variable is the sum of starboard and port propeller torque values. The GP range of GP parameters that were used in all these analyses is shown in Table 3.

**Table 3.** The range of GP parameters used in all analyses.

| GP Parameter | Lower Bound | Upper Bound |
|---|---|---|
| Population size | 500 | 1000 |
| Number of generations | 100 | 500 |
| Tournament selection size | 50 | 100 |
| Tree depth | (3–7) | (6–12) |
| Crossover coefficient | 0.9 | 1 |
| Subtree mutation coefficient | 0.01 | 0.1 |
| Hoist mutation coefficient | 0.01 | 0.1 |
| Point mutation coefficient | 0.01 | 0.1 |
| Stopping criteria value | $1 \times 10^{-6}$ | 0.001 |
| Maximum number of samples | 0.9 | 1.0 |
| Constant range | −0.1 | 0.1 |
| Parsimony coefficient | $1 \times 10^{-4}$ | 0.01 |

As seen in Table 3 the dominating genetic operator is crossover coefficient when compared to three mutation coefficient. The stopping criteria range is very small; however, in all GP algorithm execution, this value was never achieved so the GP algorithm execution was terminated when the maximum number generation was reached. The parsimony coefficient value is responsible for penalizing the large growth of population members without improvement in fitness value i.e., bloat phenomenon. The values of the parsimony coefficient in all analyses were small to allow the growth of population members from generation to generation.

### 3.1.1. The Symbolic Expressions for Fuel Flow Estimation with and without Decay State Coefficients

To obtain symbolic expressions for fuel flow estimation with decay state coefficients total of 16 input variables were used from the training dataset part and fuel flow was used as the output variable which is shown in Table 2. In the case of fuel flow estimation without decay state coefficients only 14 input variables were used. After multiple GP algorithm executions, the three best symbolic expressions with and without decay state coefficients were selected based on their performance in terms of $R^2$ and $MAE$ values, respectively. The three best symbolic expressions with and without decay state coefficients for fuel flow estimation are presented in Tables 4 and 5.

**Table 4.** Three best symbolic expressions for fuel flow estimation with decay state coefficients with corresponding $R^2$ and *MAE* score.

| GP Parameters - Population, Generations, Selection Size, Tree Depth, Crossover Coef., Subtree Mutation Coef., Hoist Mutation Coef., Point Mutation Coef., Stopping Criteria, Samples, Constant Range, Parsimony Coef. | Symbolic Expression | $R^2$ | *MAE* |
|---|---|---|---|
| [930, 243, 81, (3, 11), 0.91, 0.021, 0.015, 0.041, 0.0002, 0.95, (−0.043, 0.021), 0.0003] | $y_{mfDF1} = (\log(\min(\sqrt{\sin(\log(\frac{X_{12}}{X_{15}X_{16}}))},$ $\tan(\sin(\tan(\sin(\log(\frac{X_{12}}{X_{13}X_{15}}))))))))^{\frac{1}{2}}$ | 0.99398 | 0.02664 |
| [742, 103, 92, (4, 11), 0.9, 0.026, 0.035, 0.02, 0.0002, 0.91, (−0.071, 0.02), 0.0038] | $y_{mfDF2} = \log(X_{10})\cos(\log(\cos(X_{16})))$ $\cos(\log(\tan(X_{11})))\max(X_{15}, \log(X_{10}))$ | 0.993 | 0.03695 |
| [927, 346, 80, (6, 9), 0.9, 0.032, 0.039, 0.019, 0.0002, 0.92, (−0.063, 0.056), 0.0008] | $y_{mfDF3} = \log\left(X_{10}X_{15}\cos\left(\frac{X_{13}+\sin(X_{16}+X_3)}{\sin(X_0)+3.35241}\right)\right)$ | 0.95526 | 0.08184 |

**Table 5.** Three best symbolic expressions for fuel flow estimation without decay state coefficients with corresponding $R^2$ and *MAE* score.

| GP Parameters - Population, Generations, Selection Size, Tree Depth, Crossover Coef., Subtree Mutation Coef., Hoist Mutation Coef., Point Mutation Coef., Stopping Criteria, Samples, Constant Range, Parsimony Coef. | Symbolic Expression | $R^2$ | *MAE* |
|---|---|---|---|
| [962, 289, 52, (6, 8), 0.91, 0.017, 0.035, 0.03, 0.000524, 0.99, (−0.073, 0.0014), 0.0029] | $y_{mf1} = \dfrac{X_{10}}{\sqrt{\dfrac{\ln(X_2)\sqrt{\frac{\ln(X_2)}{\ln(X_{10})}}}{X_{10}}}}$ | 0.9964 | 0.02276 |
| [1000, 141, 83, (5, 9), 0.9, 0.022, 0.012, 0.032, 0.000986, 0.98, (−0.049, 0.0943), 0.0013] | $y_{mf2} = \sqrt{\tan(X_1)}\sin(\sqrt{\tan(\max(X_1, \ln(X_4)))}$ $\sin(\sin(\sin((\sin(\sin(\sin(\sqrt{\sin(X_1)}))))$ $\sqrt{\tan(\max(X_1, \ln(X_4)))^{\frac{1}{2}}}))))$ | 0.99591 | 0.02341 |
| [582, 365, 85, (4, 7), 0.9, 0.022, 0.027, 0.018, 0.00046, 0.91, (−0.0103, 0.0905), 0.0003] | $y_{mf3} = \dfrac{\ln(X_{10})}{\tan\left(\sin\left(\frac{\frac{\ln(X_{10})}{X_{13}}}{X_{13}}+X_{11}\right)\right)}$ | 0.99578 | 0.023027 |

When Tables 4 and 5 are compared it can be noticed that decay state coefficients are decreasing the performance of fuel flow estimation in terms of $R^2$ and $MAE$ values. The population size for each case was near 1000 except for the third case without decay state coefficients where population size is near the lower boundary of 500. The crossover coefficient was the dominating genetic operation for each case. All six symbolic expressions are small in size so the bloat phenomenon did not occur although the values of the parsimony coefficients in all six cases are extremely small. The fuel flow estimation performance of all six symbolic expressions is shown in Figure 7.

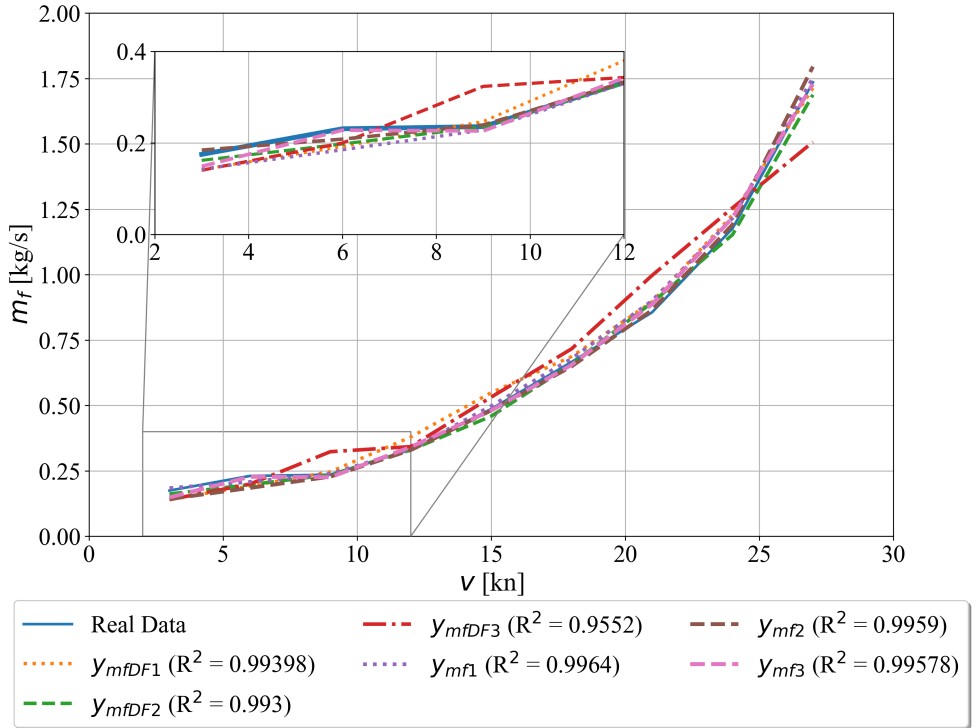

**Figure 7.** The comparison of estimated fuel flow with real data versus the ship speed.

As seen in Figure 7 all symbolic expressions are estimating the fuel flow with high accuracy except for $y_{mfDF3}$ which has the highest deviation from the real data. When the estimation performance of symbolic expressions with decay state coefficients is compared to those without decay state coefficients it can be noticed that those symbolic expressions with decay state coefficients have slightly lower estimation accuracy. However, those symbolic expressions with decay state coefficients are more important symbolic expressions for CBM since they could indicate the potential degradation of system performance.

### 3.1.2. The Symbolic Expressions for Ship Speed Estimation with and without Decay State Coefficients

In the case of ship speed estimation using GP with decay state coefficients, the total of 16 input variables was considered while in the case without decay state coefficients the GCDSC and GTDSC input variables were omitted. The output variable in both cases was the shipping speed as indicated in Table 2. After multiple GP algorithm executions using the training dataset part, all symbolic expressions were tested on the testing dataset part to determine $R^2$ and $MAE$ value. Based on the highest $R^2$ and $MAE$ value the three best symbolic expressions with and without decay state coefficients were chosen and shown in Tables 6 and 7.

**Table 6.** Three best symbolic expressions for ship speed estimation with decay state coefficients with corresponding $R^2$ and *MAE* score.

| GP Parameters - Population, Generations, Selection Size, Tree Depth, Crossover Coef., Subtree Mutation Coef., Hoist Mutation Coef., Point Mutation Coef., Stopping Criteria, Samples, Constant Range, Parsimony Coef. | Symbolic Expression | $R^2$ | *MAE* |
|---|---|---|---|
| [548, 311, 87, (3, 8), 0.91, 0.017, 0.017, 0.018, 0.000926, 0.92, (−0.015, 0.044), 0.0013] | $y_{ssDF1} = (X_{15} + X_{16})\left(\frac{X_0}{X_{10}+X_{12}} + X_0\right)$ | 0.99843 | 0.2858 |
| [784, 458, 77, (4, 7), 0.9, 0.015, 0.015, 0.06, $9.3 \times 10^{-5}$, 0.9, (−0.0083, 0.082), 0.0063] | $y_{ssDF2} = X_0 X_{15} X_{16} + X_0 X_{15} + X_0 X_{16}$ | 0.99788 | 0.32584 |
| [585, 286, 69, (3, 12), 0.9, 0.024, 0.025, 0.023, 0.000191, 0.92, (−0.00084, 0.018), 0.0053] | $y_{ssDF3} = \|\|\log(X_{14})\| + \tan(X_{15} + X_{16})\| + \sqrt{X_4}$ | 0.99593 | 0.41067 |

**Table 7.** Three best symbolic expressions for ship speed estimation without decay state coefficients with corresponding $R^2$ and *MAE* score.

| GP Parameters - Population, Generations, Selection Size, Tree Depth, Crossover Coef., Subtree Mutation Coef., Hoist Mutation Coef., Point Mutation Coef., Stopping Criteria, Samples, Constant Range, Parsimony Coef. | Symbolic Expression | $R^2$ | *MAE* |
|---|---|---|---|
| [732, 352, 86, (6, 10), 0.92, 0.012, 0.013, 0.023, 0.000231, 0.9, (−0.073, 0.031), 0.003] | $y_{sp1} = \frac{X_0 - 0.066}{X_{12}} + 2X_0 - 0.279$ | 0.9998925 | 0.06729 |
| [945, 479, 70, (6, 7), 0.91, 0.016, 0.016, 0.014, $9.4 \times 10^{-5}$, 0.98, (−0.085, 0.0049), 0.0097] | $y_{sp2} = \sqrt{X_0(X_0 - X_{14})\log\left(X_3 + X_4\sqrt{X_6}\right)}$ | 0.999825 | 0.08665 |
| [690, 152, 82, (6, 12), 0.9, 0.047, 0.01, 0.018, $3.6 \times 10^{-5}$, 0.94, (−0.023, 0.058), 0.0078] | $y_{sp3} = X_{14}\cos(X_{12} - X_{14}\cos(X_0 - X_{10})) + \log(X_0) + \sqrt{X_4}$ | 0.999541 | 0.11797 |

As seen in Tables 6 and 7 those symbolic expressions with decay state coefficients included in the analyses have slightly lower estimation accuracy in terms of $R^2$ and $MAE$ values when compared to those symbolic expressions without decay state coefficients. As in the case of fuel flow estimations, both decay state coefficients are in all three symbolic expressions shown in Table 6. In this analysis, the crossover coefficient was the dominating genetic operation when compared to the remaining three mutation coefficient values, and the parsimony coefficient was extremely low. The tree depth range of the initial population was lower in the case of symbolic expressions with decay state coefficients. The stopping criteria value in all these analyses was never achieved due to the extremely low value, so the GP algorithm executions were terminated after the maximum number of generations was reached. The estimation performance of all six symbolic expressions is shown in Figure 8.

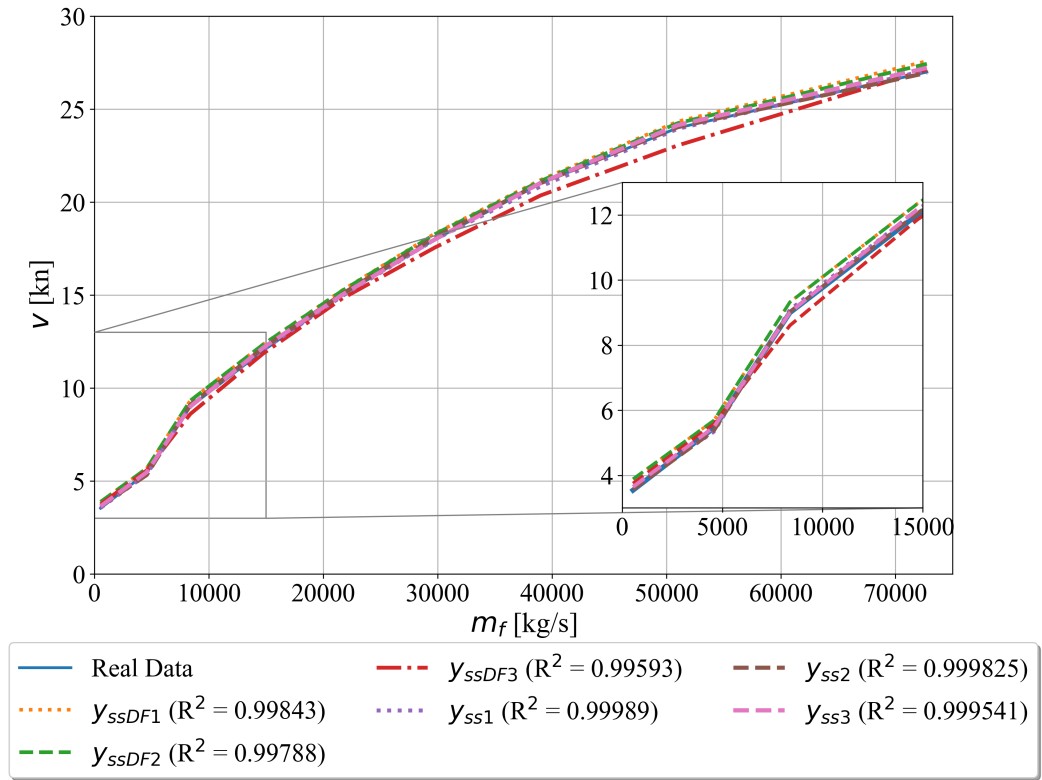

**Figure 8.** The Comparison of Estimated Ship Speed with Real Data Versus the Fuel Flow.

In Figure 8 the variation of ship speed versus the fuel flow is shown. The estimation accuracy of ship speed using symbolic expressions with decay state coefficients is slightly lower than those without decay state coefficients which are also indicated by achieved $R^2$ and $MAE$ values.

### 3.1.3. The Symbolic Expressions for Starboard Propeller Torque Estimation with and without Decay State Coefficients

In the case of starboard propeller torque analysis using the GP algorithm, the port propeller torque was excluded from the analysis since it has almost identical values as the starboard propeller torque. Therefore, if the port propeller torque was included as an input variable in the GP algorithm this would result in early termination of GP algorithm execution. With the exclusion of port propeller torque from the analysis, the total number of input variables in the case of decay state coefficient is 15 while in the case without decay state coefficient the total number of input variables is 13. The list of input and output variables is shown in Table 2. After multiple GP algorithm executions using the training dataset part the obtained symbolic expressions were evaluated on the testing dataset part

to determine the $R^2$ and $MAE$ values, respectively. Based on the highest $R^2$ and lowest $MAE$ values the three best symbolic expressions with and without decay state coefficients were selected and shown in Tables 8 and 9 with corresponding GP parameters.

**Table 8.** Three best symbolic expressions for starboard torque estimation with decay state coefficients with corresponding $R^2$ and $MAE$ score.

| GP Parameters - Population, Generations, Selection Size, Tree Depth, Crossover Coef., Subtree Mutation Coef., Hoist Mutation Coef., Point Mutation Coef., Stopping Criteria, Samples, Constant Range, Parsimony Coef. | Symbolic Expression | $R^2$ | $MAE$ |
|---|---|---|---|
| [996, 399, 69, (3, 10), 0.92, 0.013, 0.035, 0.018, $9.06 \times 10^{-7}$, 0.94, ($-0.078$, 0.077), 0.0061] | $y_{stDF1} = X_0 + X_1 X_{10} + 5X_{13} + X_8 + X_{SPTDF11} + X_{SPTDF12}$ | 0.99985 | 1.98477 |
| [821, 418, 92, (4, 10), 0.909, 0.044, 0.018, 0.011, $1.46 \times 10^{-7}$, 0.98, ($-0.002$, 0.07), 0.0022] | $y_{stDF2} = X_1 X_{10} \min(X_{11}, X_{SPTDF21})$ | 0.99959 | 3.16776 |
| [598, 398, 63, (4, 11), 0.9, 0.033, 0.018, 0.032, $1.68 \times 10^{-7}$, 0.96, ($-0.0055$, 0.014), 0.0016] | $y_{stDF3} = X_{12} X_{SPTDF31}$ | 0.99737 | 7.9579 |

**Table 9.** Three best symbolic expressions for starboard torque estimation without decay state coefficients with corresponding $R^2$ and $MAE$ score.

| GP Parameters - Population, Generations, Selection Size, Tree Depth, Crossover Coef., Subtree Mutation Coef., Hoist Mutation Coef., Point Mutation Coef., Stopping Criteria, Samples, Constant Range, Parsimony Coef. | Symbolic Expression | $R^2$ | $MAE$ |
|---|---|---|---|
| [554, 233, 81, (5, 11), 0.9, 0.052, 0.025, 0.017, $5.1 \times 10^{-7}$, 0.95, ($-0.087$, 0.028), 0.0031] | $y_{st1} = \sqrt{X_{SPT11} X_{SPT12}}$ | 0.99994 | 1.0697 |
| [792, 144, 63, (5, 8), 0.92, 0.039, 0.013, 0.025, $6.25 \times 10^{-7}$, 0.92, ($-0.07$, 0.01), 0.0069] | $y_{st2} = X_0(X_1 + X_{SPT21})$ | 0.99989 | 1.3387 |
| [824, 297, 57, (6, 7), 0.91, 0.014, 0.032, 0.031, $4.08 \times 10^{-7}$, 0.92, ($-0.08$, 0.039), 0.0039] | $y_{st3} = \frac{X_1 X_{10} X_{SPT31}}{\tan(\tan(X_9))} + X_1 X_{10} + \log(X_{13}) + X_{SPT32}$ | 0.99981 | 1.8535 |

As seen in Tables 8 and 9 some new variables were introduced to shorten the size of symbolic expressions in the aforementioned tables. The full form of $X_{SPTDF11}$, $X_{SPTDF12}$, $X_{SPTDF21}$, $X_{SPTDF31}$, $X_{SPT11}$, $X_{SPT12}$, $X_{SPT21}$, $X_{SPT31}$, and $X_{SPT32}$ is shown in Appendices A.1 and A.2, respectively. The $R^2$ values of symbolic expressions with decay state coefficients in the estimation of starboard propeller torque are slightly lower when compared to the symbolic expressions without decay state coefficients while the $MAE$ values are higher in symbolic expressions with decay state coefficients when compared to the symbolic expressions obtained without decay state coefficients. The stopping criteria values in all six symbolic expressions are extremely low when compared to the fuel flow and ship speed analysis. Again, these values were never achieved so the GP execution was terminated after a maximum number of generations was reached. The values of the parsimony coefficient were low in all six symbolic expressions which generated very large symbolic expressions so the aforementioned coefficients were introduced to simplify their form. The other key factor that contributed to large symbolic expressions is the constants range which in all analyses is very low. Therefore, the GP algorithm had to replace the low constants range by increasing the size of symbolic expressions using mathematical functions. The estimation performance of starboard propeller torque with and without decay state coefficients compared to real data are shown in Figure 9.

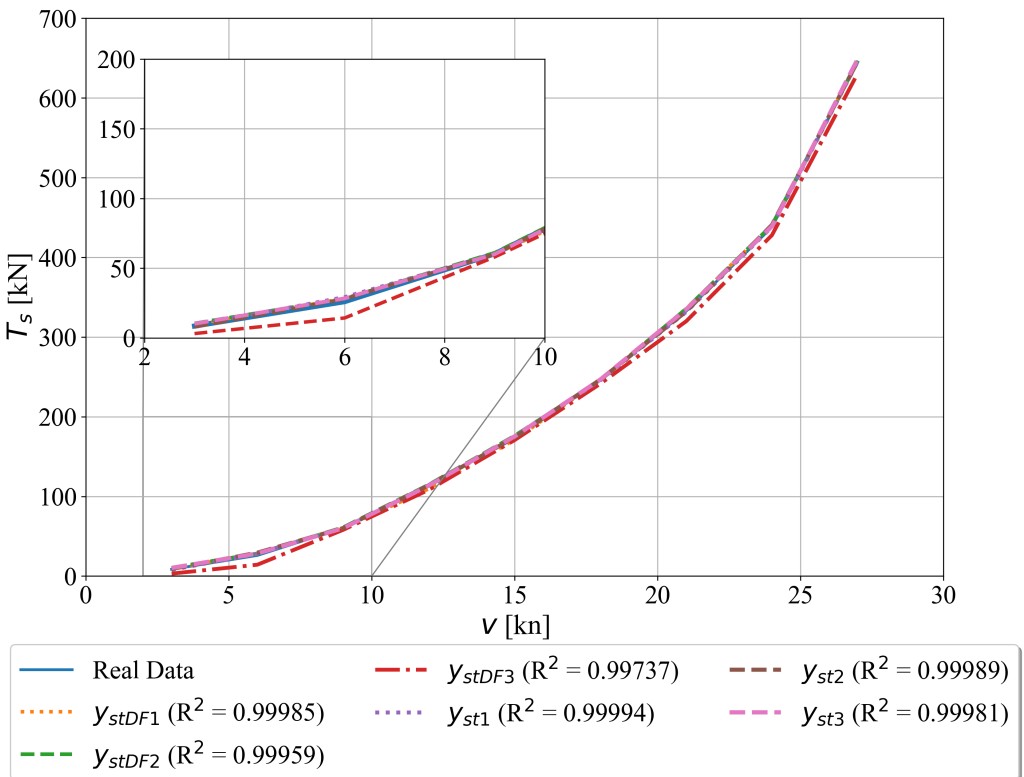

**Figure 9.** The Variation of Real and Estimated Starboard Propeller Torque Values versus Ship Speed.

In Figure 9, it can be noticed that all symbolic expressions have an accurate estimation of starboard propeller torque when compared to the values from the dataset. However, the third symbolic expressions with decay state coefficients have the lowest estimation accuracy when compared to the remaining five which can also be indicated with a lower $R^2$ value or higher $MAE$ value, respectively.

### 3.1.4. The Symbolic Expressions for Port Propeller Torque Estimation with and without Decay State Coefficients

The procedure of obtaining symbolic expressions for estimation of port propeller torque with and without decay state coefficient is similar to the procedure of obtaining the symbolic expressions for starboard propeller torque. The starboard propeller torque was omitted as an input variable from the investigation due to the equal values as port propeller torque. Initial investigation of port propeller torque using GP algorithm with the inclusion of starboard propeller torque showed early termination of GP algorithm. In the case of symbolic expressions with decay state coefficient included there was a total of 15 input variables while in the case without decay state coefficients there was a total of 13 input variables, while the port propeller torque was output variable. The list of input and output variables is shown in Table 2. The equations are not based on previous knowledge or derived from other findings-but generated purely through the evolutionary process of GP described in the Methodology, which attempts to, in a heuristic manner, develop equations that provide a high fitness value for the used dataset. After multiple executions with the GP algorithm using the training dataset part the obtained symbolic expressions were evaluated on the testing dataset part to determine the $R^2$ and $MAE$ value. Based on the highest $R^2$ value and lowest $MAE$ values the three best symbolic expressions with and without decay state coefficients were chosen and shown in Tables 10 and 11.

**Table 10.** Three best symbolic expressions for port propeller torque estimation with decay state coefficients with corresponding $R^2$ and $MAE$ score.

| GP Parameters - Population, Generations, Selection Size, Tree Depth, Crossover Coef., Subtree Mutation Coef., Hoist Mutation Coef., Point Mutation Coef., Stopping Criteria, Samples, Constant Range, Parsimony Coef. | Symbolic Expression | $R^2$ | $MAE$ |
|---|---|---|---|
| $[788, 470, 94, (6, 8),$ $0.93, 0.016, 0.017, 0.032,$ $6.00 \times 10^{-9}, 0.93,$ $(-0.072, 0.083), 0.0044]$ | $y_{pptDF1} = \dfrac{(\log(\log(X_0)) + X_{12})}{\log(X_9 - X_0) + X_{PPTDF11}}$ | 0.99964 | 1.9885 |
| $[979, 263, 77, (6, 8),$ $0.91, 0.047, 0.012, 0.022,$ $6.86 \times 10^{-9}, 0.91,$ $(-0.062, 0.0027), 0.0043]$ | $y_{pptDF2} = X_1 X_{10} + X_{PPTDF21}$ | 0.9996 | 2.61963 |
| $[986, 394, 53, (3, 12),$ $0.91, 0.018, 0.051, 0.013,$ $9.47 \times 10^{-9}, 0.946,$ $(-0.02, 0.016), 0.0095]$ | $y_{pptDF3} = X_0 X_{PPTDF31}$ | 0.99427 | 14.0996 |

**Table 11.** Three best symbolic expressions for port propeller torque estimation without decay state coefficients with corresponding $R^2$ and $MAE$ score.

| GP Parameters - Population, Generations, Selection Size, Tree Depth, Crossover Coef., Subtree Mutation Coef., Hoist Mutation Coef., Point Mutation Coef., Stopping Criteria, Samples, Constant Range, Parsimony Coef. | Symbolic Expression | $R^2$ | $MAE$ |
|---|---|---|---|
| [709, 445, 70, (5, 11), 0.9, 0.023, 0.029, 0.041, $5.72 \times 10^{-7}$, 0.94, $(-0.041, 0.01)$, 0.0031] | $y_{ppt_1} = \frac{X_0 X_{11} X_{PPT11}}{X_{11} + X_9}$ | 0.9994 | 3.35254 |
| [986, 294, 74, (4, 12), 0.91, 0.042, 0.012, 0.025, $6.99 \times 10^{-7}$, 0.91, $(-0.021, 0.081)$, 0.0061 ] | $y_{ppt_2} = X_{10}(\min(X_{13}, \log(|X_{PPT21}|)) + X_1 + 0.276)$ | 0.99922 | 4.06154 |
| [769, 415, 69, (3, 11), 0.93, 0.014, 0.011, 0.028, $4.21 \times 10^{-7}$, 0.96, $(-0.067, 0.035)$, 0.0046] | $y_{ppt_3} = (X_1 + X_{13})\left( \frac{X_{12} \sin(X_0)\sqrt{\sin^3\left(\sin\left(\sqrt{X_{12}}\right)\right) X_{PPT31}}}{X_{10}} + X_{PPT32} \right)^{\frac{1}{2}}$ | 0.99891 | 5.11714 |

Due to the large size of obtained symbolic expressions the coefficients $X_{PPTDF11}$, $X_{PPTDF21}$, $X_{PPTDF31}$, $X_{PPT11}$, $X_{PPT21}$, $X_{PPT31}$, and $X_{PPT32}$. The full form of these coefficients is given Appendices A.3 and A.4. Although the parsimony coefficient value for all symbolic expressions is low the bloat phenomenon did not occur. However, the large size of obtained symbolic expressions could be explained by the low range of constant values. Since this range is very low the GP algorithm used a large number of mathematical functions and input variables to achieve high estimation accuracy. Based on $R^2$ and $MAE$ values the symbolic expressions with and without decay state coefficients have almost similar performance except for the third symbolic expression which has the lowest $R^2$ value and highest $MAE$ value. The estimation performance of these six symbolic expressions are compared to the real data and shown in Figure 10.

The estimation performance of all six symbolic expressions is very high when compared to the real data except for the third symbolic expression with decay state coefficient which performed poorly when compared to the other symbolic expressions.

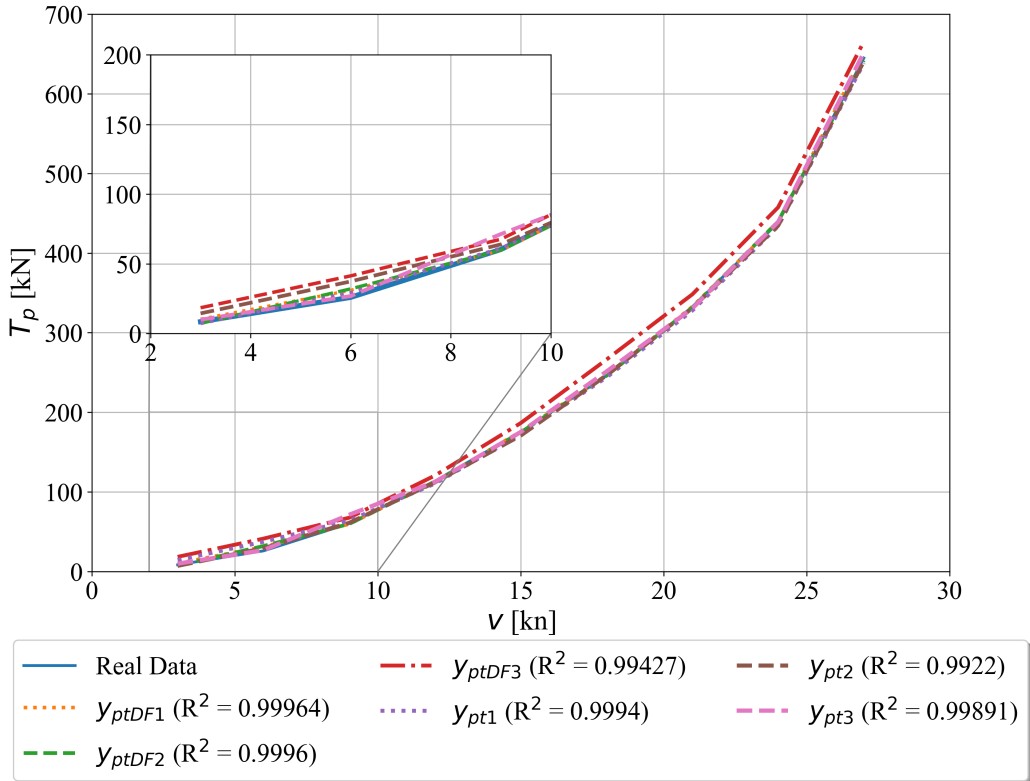

**Figure 10.** The variation of real and estimated port propeller torque versus the ship speed.

3.1.5. The Symbolic Expressions for Total Propeller Torque Estimation with and without Decay State Coefficients

To obtain symbolic expressions for total propeller torque estimation the starboard and port propeller torque were added together. This variable was used as the output variable in the training and testing portion of the dataset. The starboard and port propeller torque as input variables were omitted from the analysis so the total number of variables was 15 in the case where decay state coefficients were used and 13 in the case without decay state coefficients. After multiple GP executions using the training portion of the dataset the obtained symbolic expressions were evaluated on the testing portion of the dataset to determine $R^2$ and $MAE$ value. Based on the highest $R^2$ and lowest $MAE$ value the best symbolic expressions with and without decay state coefficients are chosen. The symbolic expressions with and without decay state coefficients are shown in Tables 12 and 13.

In Table 12 each symbolic expression has at least one decay state coefficient since the GP algorithm could not obtain the symbolic expression for estimation of total torque with both decay state coefficients. To simplify presentation of symbolic expressions in Tables 12 and 13 coefficients $X_{TTDF11}$, $X_{TT11}$, $X_{TT12}$, $X_{TT21}$, $X_{TT31}$, and $X_{TT32}$ were introduced. The full form of these coefficient is given in Appendices A.5 and A.6. The $R^2$ values of symbolic expressions with decay state coefficients are lower while $MAE$ values are higher than those values obtained using symbolic expressions without decay state coefficients. The graphical representation and estimation performance of six symbolic expressions from Tables 12 and 13 are shown in Figure 11.

**Table 12.** Three best symbolic expressions for total propeller torque estimation with decay state coefficients with corresponding $R^2$ and *MAE* score.

| GP Parameters - Population, Generations, Selection Size, Tree Depth, Crossover Coef., Subtree Mutation Coef., Hoist Mutation Coef., Point Mutation Coef., Stopping Criteria, Samples, Constant Range, Parsimony Coef. | Symbolic Expression | $R^2$ | *MAE* |
|---|---|---|---|
| [910, 285, 69, (5, 9), 0.91, 0.038, 0.016, 0.015, $8.42 \times 10^{-7}$, 0.94, $(-0.029, 0.09)$, 0.0096] | $y_{ttDF1} = \|X_{TTDF11}\| + X_{12}$ $\sin(\sin(\sin(\log(\tan(\sin(\sqrt{X_0})) - X_0)))))$ | 0.99848 | 11.697387 |
| [664, 116, 75, (3, 10), 0.9, 0.058, 0.015, 0.017, $3.4e \times 10^{-7}$, 0.93, $(-0.048, 0.015)$, 0.0096] | $y_{ttDF2} = \min(X_{13}, X_{14}) \max(X_5, X_{13}X_7)$ $-\sqrt{\max(X_5, X_{13}^2 X_7) - 2\tan(\sqrt{X_3})}$ | 0.991606 | 26.33334 |
| [790, 112, 79, (3, 12), 0.91, 0.012, 0.031, 0.021, $7.67 \times 10^{-7}$, 0.9, $(-0.02, 0.054)$, 0.007] | $y_{ttDF3} = X_{13}X_{15}\min(X_5, X_7)$ | 0.97971 | 49.89208 |

**Table 13.** Three best symbolic expressions for total propeller torque estimation without decay state coefficients with corresponding $R^2$ and *MAE* score.

| GP Parameters - Population, Generations, Selection Size, Tree Depth, Crossover Coef., Subtree Mutation Coef., Hoist Mutation Coef., Point Mutation Coef., Stopping Criteria, Samples, Constant Range, Parsimony Coef. | Symbolic Expression | $R^2$ | *MAE* |
|---|---|---|---|
| [682, 172, 56, (4, 7), 0.9, 0.018, 0.025, 0.029, $4.09 \times 10^{-7}$, 0.93, $(-0.012, 0.065)$, 0.0026] | $y_{tt1} = \|X_{12} - X_{TT11}\|$ $-X_{TT12} - \sqrt{\frac{X_3}{X_8}} + X_6 X_8$ | 0.99808 | 9.2407 |
| [798, 103, 77, (4, 11), 0.9, 0.01, 0.061, 0.021, $6.14 \times 10^{-7}$, 0.92, $(-0.039, 0.046)$, 0.0069] | $y_{tt2} = \frac{X_{12}X_8 + \sqrt{X_2}}{X_{TT21}}$ | 0.99806 | 13.25 |
| [883, 209, 64, (6, 9), 0.93, 0.013, 0.023, 0.028, $8.75 \times 10^{-7}$, 0.96, $(-0.057, 0.057)$, 0.0099] | $y_{tt3} = \max\left(\frac{X_{TT31}}{X_{TT32}} + \sqrt{X_3}, \log(X_2) - X_{12}\right) + X_{12}$ | 0.9976 | 13.6284 |

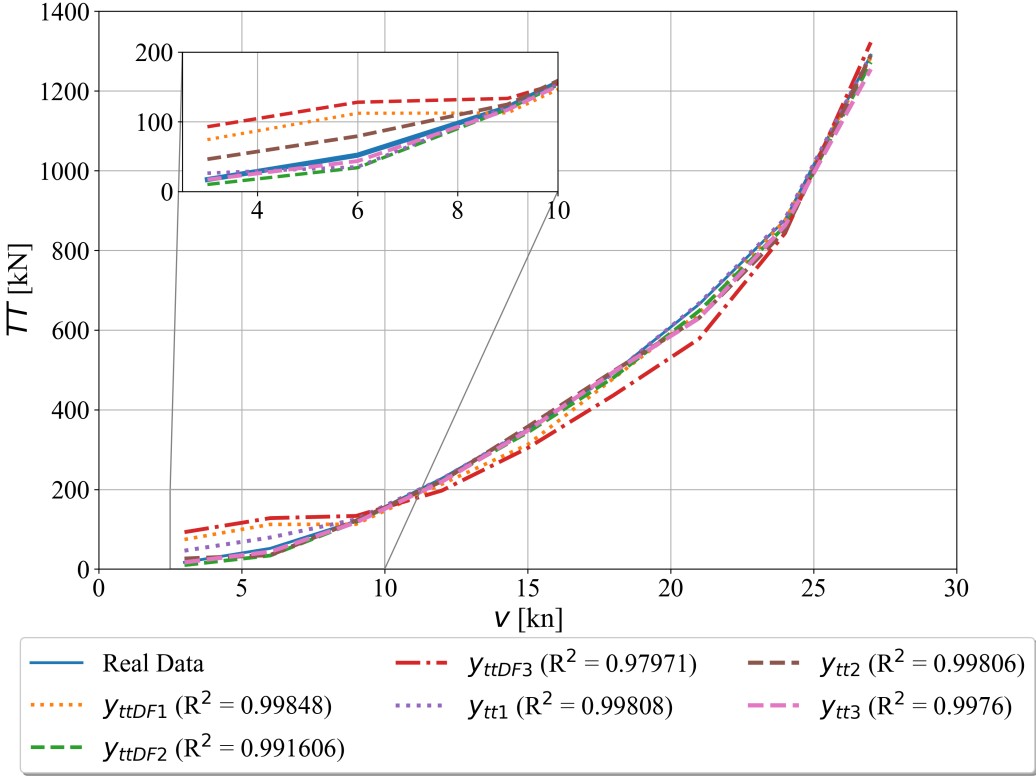

**Figure 11.** The variation of real and estimated total torque versus ship speed.

As seen in Figure 11 the $y_{ttDF1}$, $y_{ttDF3}$ and $y_{tt1}$ have some deviation from the real data at lower ship speeds. However, the highest deviation from the real data through entire ship speed range is produced by $y_{ttDF3}$.

### 3.2. Discussion

From conducted investigation, it can be noted that two correlation analyses showed that 14 out of 18 dataset variables (without decay state coefficients, T1, and P1) have positive correlation values with remaining variables in the range from 0.8791 up to 1.0. The T1 and P1 showed no correlation with any other variable except with itself. The reason why these two variables do not correlate is that they are constant values through the entire dataset as seen from Table 1. As already stated these two variables represent ambient temperature and pressure which were constant during the simulation of the CODLAG propulsion system. The turbo compressor and turbine decay state coefficients have positive, negative, or no correlation with other variables in the dataset. The analysis showed that with ship speed two decay state coefficients do not have any correlation at all since the correlation values are equal to zero. The Pearson's correlation analysis showed that two decay state coefficients have a small positive correlation (0.0008, 0.0001) with starboard and port propeller torque while Spearman's correlation analysis showed that two decay state coefficients have a negative and positive correlation ($-0.0295$, 0.0174) with starboard and port propeller torque. It should be noted that the correlation with fuel flow and decay state coefficients is negative in Pearson's and Spearman's correlation analysis.

Regardless of the results from two correlation analysis, the idea was to investigate the possibility of using the GP algorithm to obtain symbolic expressions for estimation of fuel flow, ship-speed, starboard, port, and total propeller torque with and without decay state coefficients since those two coefficients are possible indicators of GT system parts degradation. The total propeller torque was generated by adding together values of starboard and port propeller torque. All symbolic expressions were obtained on the training portion of the dataset with the proper definition of input and output dataset values as indicated in Table 2 and with a random selection of GP, parameters range shown in Table 3

in each GP algorithm execution. It should be noted that in the entire investigation using the GP algorithm the crossover operation was the dominant genetic operation and that predefined (randomly selected) stopping criteria value was never achieved by any of the population members. Therefore, each execution of the GP algorithm was terminated after the maximum number of generations was reached. After the symbolic expressions were obtained they were tested on the test part of the dataset to obtain $R^2$ and $MAE$ values. The three best symbolic expressions in each case with and without decay state coefficients were chosen based on their highest $R^2$ value and the lowest $MAE$ values. Another interesting thing is that all these symbolic expressions were obtained with a minimum range of constants which means that in the majority of cases the symbolic expressions consist of mathematical expressions and input variables. Some symbolic expressions grew in size to achieve low estimation error between calculated output and desired output. However, the parsimony coefficient range was low but the bloat phenomenon did not occur.

In the case of symbolic expressions for fuel flow estimation the symbolic expressions with decay state coefficients have slightly lower $R^2$ values (0.99398, 0.993, 0.95526) and slightly higher $MAE$ (0.02664, 0.03695, 0.08184) values when compared to $R^2$ (0.9964, 0.99591, 0.99578) and $MAE$ (0.02276, 0.02341, 0.023027) values obtained using symbolic expressions without decay state coefficients. The best symbolic expression with decay state coefficients has almost similar estimation performance of fuel flow when compared to the symbolic expressions obtained without decay state coefficients. Therefore, including those two decay state coefficients resulted in slightly lower performance of obtained symbolic expressions. However, these three symbolic expressions with decay state coefficients are highly valuable since they could indicate potential degradation of the GT propulsion system in terms of higher fuel consumption without noticeable improvement in propeller torque or ship speed.

In the case of ship speed estimation the three obtained symbolic expressions with decay state coefficients have achieved lower $R^2$ (0.99843, 0.99788, and 0.99593) and higher $MAE$ (0.2858, 0.32584, and 0.41067) values when compared to $R^2$ (0.9998925, 0.999825, and 0.999541) and $MAE$ (0.06729, 0.08665, 0.11797) values achieved with symbolic expressions obtained without decay state coefficients. Interestingly, those two decay state coefficients do not influence ship speed since Pearson's and Spearman's correlation analysis showed that these two coefficients do not have any correlation with ship speed. Therefore, in the case of those three symbolic expressions obtained with decay state coefficients, the other input variables are $X_0$, $X_4$, $X_{10}$, and $X_{14}$ which are lever position, starboard propeller torque, turbo compressor inlet air pressure (P1), GT exhaust gas pressure, and fuel flow, respectively.

The estimation performance of starboard propeller torque with decay state coefficient is lower when $R^2$ (0.99985, 0.99959, and 0.99737) and $MAE$ (1.98477, 3.16776, and 7.9579) values are compared to $R^2$ (0.99994, 0.99989, and 0.99981) and $MAE$ (1.0697, 1.3387, and 1.8535) values of three symbolic expressions obtained without decay state coefficients. It should be noted that in these symbolic expressions the additional coefficients were introduced to simplify their presentation in Tables 8 and 9 while the full form of these coefficients is shown in Appendices A.1 and A.2. The correlation analysis showed that starboard propeller torque has a positive Pearsons correlation with both decay state coefficients while negative correlation coefficient with GCDSC and positive correlation with GTDSC. The symbolic expressions for estimation of port propeller torque showed similar behavior as in the case of starboard propeller torque. The values of Pearson's and Spearman correlation values of port propeller torque and decay state coefficients are the same as in the case of starboard propeller torque.

In the case of total propeller torque, the symbolic expressions with decay state coefficients achieved higher $MAE$ values than those obtained without decay state coefficients which means that the decay state coefficient contributed to higher error rates. In terms of $R^2$ values, the first symbolic expression in Table 12 achieved a similar value as those symbolic expressions obtained without decay state coefficients which are shown in Table 13.

With the use of the GP algorithm, none of the obtained symbolic expressions with decay state coefficients, including the best three symbolic expressions shown in Table 12 did not include both of the decay state coefficients. The estimation performance is lower at low ship speeds and they are increasing as the ship speed also increases. Generally, lower estimation performance can be noticed for symbolic expressions with decay state coefficients when compared to those obtained without decay state coefficients. In comparison to the previous work in the field, refs [10–12] it can be seen that GP implementation in this paper achieves comparable results to other works using it. The same can be said for other researchers with similar goals, such as [36] in which the used methods achieve results that are comparable to the ones achieved by GP. In comparison to the performance of the existing work in AI-based CODLAG system modeling, which used other ML algorithms it is seen that results achieved by GP are comparable or better, with the benefit of clearer models. The clearer models in question make it possible to see which of the inputs (such as decay coefficients) ended up not being included in the best performing models signifying their low influence in the final model.

## 4. Conclusions

In this paper, the publicly available dataset of the CODLAG propulsion system was used in the GP algorithm to obtain the symbolic expressions for fuel flow, ship speed, starboard propeller torque, port propeller torque, and total propeller torque estimation with and without decay state coefficients. From the extensively conducted investigations, the following conclusions can be drawn:

- the Pearson's and Spearman's correlation analysis showed that from a total of 18 variables in the dataset 14 of them (without decay state coefficient, T1, and P1) have positive correlation values. The turbo compressor decay state coefficient and turbine decay state coefficient do not correlate with ship speed, have positive Pearsons correlation with starboard and port propeller torque, have positive and negative Spearman's correlation with starboard and port propeller torque, and negative correlation with fuel flow. The T1 and P1 represent ambient temperature and pressure so they are constant values throughout the entire dataset. Hence there are not any correlation values with other parameters in the dataset.
- the GP algorithm can be used to obtain symbolic expressions for estimation of fuel flow, ship speed, starboard propeller torque, port propeller torque, and total propeller torque with and without decay state coefficients for the observed CODLAG propulsion system,
- the symbolic expressions for estimation of fuel flow, ship speed, starboard propeller, port propeller and total propeller torque with decay state coefficients generally have slightly lower $R^2$ and slightly higher $MAE$ values when compared to those symbolic expressions obtained without decay state coefficients. However, those symbolic expressions with decay state coefficients are more valuable from the CBM perspective which mean that they could be used to estimate or potentially predict possible degradation system states and schedule the system maintenance,
- the symbolic expressions for estimation of starboard propeller, port propeller, and total propeller torque with and without decay state coefficients showed slightly lower estimation performance for lower ship speeds.

Based on the conducted investigation, it can be concluded that the GP algorithm can be used for the estimation of CODLAG propulsion system-specific variables. The use of decay state coefficients in symbolic expressions can produce more realistic symbolic expressions which potentially could be used to predict possible performance degradation of the CODLAG propulsion system. The findings of the paper demonstrate the ability of the application of GP for the regression of the CODLAG system parameters. Academical applications are the possibility for the use of the determined equations for a precise determination of the regressed system parameters. Such an approach can greatly decrease the time necessary for the modeling of the system at various operating points. The use of GP

as opposed to different AI-based modeling techniques is the shape of the generated models, which are mathematical equations, that can be easily and more simply implemented within existing or newly developed systems as they are not limited to an individual programming language or a specific library as is commonly the case. While only the CODLAG system, in particular, is modeled, the approach may be applied to different propulsion systems for which the data can be collected in future work.

**Author Contributions:** Conceptualization, N.A., I.P., V.M. and Z.C.; methodology, N.A., S.B.Š., I.L. and V.M.; software, N.A., S.B.Š. and I.L.; validation, I.P., V.M. and Z.C.; formal analysis, S.B.Š., I.L. and I.P.; investigation, N.A. and I.L.; resources, N.A., S.B.Š. and I.L.; data curation, S.B.Š., I.L., I.P. and V.M.; writing—original draft preparation, N.A., S.B.Š. and I.L.; writing—review and editing, I.P., V.M. and Z.C.; visualization, N.A. and V.M.; supervision, I.P. and Z.C.; project administration, V.M. and Z.C.; funding acquisition, V.M. and Z.C. All authors have read and agreed to the submitted version of the manuscript.

**Funding:** This research received no external funding.

**Data Availability Statement:** The study used a publicly available dataset obtainable at: https:// archive.ics.uci.edu/ml/datasets/Condition+Based+Maintenance+of+Naval+Propulsion+Plants (accessed on 25 April 2021).

**Acknowledgments:** This research has been supported by the Croatian Science Foundation under the project IP- 2018-01-3739, CEEPUS network CIII-HR-0108, European Regional Development Fund under the grant KK.01.1.1.01.0009 (DATACROSS), project CEKOM under the grant KK.01.2.2.03.0004, CEI project "COVIDAi" (305.6019-20), University of Rijeka scientific grants: uniri-tehnic-18-275-1447, uniri-tehnic-18-18-1146 and uniri-tehnic-18-14.

**Conflicts of Interest:** The authors declare no conflict of interest.

## Appendix A. Coefficients in Symbolic Expressions

The coefficient of symbolic expressions that are defined for estimation of starboard propeller torque, port propeller torque, and total propeller torque with and without decay state coefficients are given. It should be noted that the GP differently treats division, natural logarithm, and square root function during its execution to avoid infinite values and complex numbers. The division function:

$$y_{DIV}(x_1, x_2) = \begin{cases} \frac{x_1}{x_2} \text{ if } |x_2| > 0.001 \\ \frac{x_1}{x_2} = 1 \text{ if } x_2 = 0 \end{cases}. \tag{A1}$$

The natural logarithm function:

$$y_{LOG}(x_1) = \begin{cases} \log(|x_1|) \text{ if } |x_1| > 0.001 \\ \log(x_1) = 0 \text{ otherwise} \end{cases}. \tag{A2}$$

The square root function:

$$y_{SQRT}(x_1) = \sqrt{|x_1|}, \tag{A3}$$

The variables $x_1$ and $x_2$ do not have any connections with input variables that were used in symbolic expressions since they are general variable names used as arguments in previously defined functions.

*Appendix A.1. Coefficients in Symbolic Expressions for Starboard Propeller Torque Estimation with Decay State Coefficients*

$$\begin{aligned} X_{SPTDF11} = {} & \min\left(X_{13} - 3X_9, X_0\left(X_{15} - \sqrt{X_8}\right)\tan(\tan(-\tan(X_0) + X_{13} - 2X_{15}))\right) + \\ & \min(X_{13} - X_9, \tan(X_0)) + \min(X_0, \min(\cos(X_0) - 2\tan(X_0) - X_9, \\ & \tan(X_0)) - \tan(\tan(X_0)) - \tan(X_{13} - 2X_9) - X_9) \end{aligned} \tag{A4}$$

$$X_{SPTDF12} = X_0 \cos(X_{10})\left(X_9 - \sqrt{X_8}\right) + \left(X_0 - \sqrt{X_8}\right)\cos(X_{10})\left(X_9 - \sqrt{X_8}\right) \tag{A5}$$

$$
\begin{aligned}
X_{SPTDF21} = {}& \sin(\sin((\min(X_0, X_{12}) + \sin(\min(X_0, X_{12}) + \sin(\sin(\sin(\min(X_0, X_{12}) + \\
& \sin(\min(X_0, X_{12}) + X_0) + \sin(\sin(\min(X_0, X_{12}) + \sin(X_0))) + X_0)))) + \\
& |\cos(X_{10})| + X_0 + \sin(\sin(X_0)))^{\frac{1}{2}} + \min(X_0, X_{12}) + \sin(\sin(2X_0)))) + \\
& \min(X_0, X_{12}, X_8, \tan(X_{15}))
\end{aligned}
\tag{A6}
$$

$$
\begin{aligned}
X_{SPTDF31} = {}& \min(X_1 \cos(X_9), X_1 \cos(X_9)\log(X_1 \cos(X_6)\log(X_1 \cos^2(X_9))) \\
& \log(X_1 \cos(X_9)\cos(\sqrt{\min(X_{11}, X_{14})})), \log(X_6 \log(X_6 \cos(\log(X_6 \\
& \log(X_6 \log(\cos(X_8 X_9)))\log(\cos(X_6)\cos(X_9)\sin(\tfrac{X_7}{X_6}))))))))
\end{aligned}
\tag{A7}
$$

*Appendix A.2. Coefficients in Symbolic Expression for Starboard Propeller Torque Estimation without Decay State*

$$
\begin{aligned}
X_{SPT11} = {}& \left(\max(\log(X_3 - \frac{\min(X_{12}, X_2^4 \tan(X_0))}{X_{12}}), -|\sin(X_8)| \right. \\
& \left. - \tan(X_0)\sin(\sqrt{\tan(X_0) - X_1}) + X_1 + \sin(X_1) + X_{12} - X_9\right)^{\frac{1}{2}}
\end{aligned}
\tag{A8}
$$

$$
\begin{aligned}
X_{SPT12} = {}& \min(X_5 \cos(\log(X_1)), \min(X_{12}, X_6) - \left(\min(\sqrt{X_8}\cos(\log(X_1)), \frac{\log(X_4)}{X_2}) + \right. \\
& \left. \frac{\log(X_4)}{X_2}\right)^{\frac{1}{2}} |\frac{X_2 X_8}{\cos(X_{11}) - \log(X_3 + X_2 X_8 - \frac{\min(X_2^2 X_5 \sin(X_1), X_5 \sin(\sin(\sin(\sin(X_1)))))}{\min(X_{12}, \frac{\log(X_4)}{X_2})})}|)
\end{aligned}
\tag{A9}
$$

$$X_{SPT21} = \frac{\max(X_0 X_8, \frac{X_1}{X_9^{11}} + X_{13} + \sin(X_3) + 2\sin(\log(X_3)))}{X_9^3} \tag{A10}$$

$$
\begin{aligned}
X_{SPT31} = {}& \sin\left(\left(\max(-0.057\sin(\sqrt{X_1 X_{10}})\csc(\sin(\sin(\sin(\sqrt{X_2})))))(\log(\min(X_{12}, \right.\right. \\
& \sqrt{\min(X_7, X_{11} + \tan(\tfrac{X_3}{X_7}))}))) + \sin(\frac{X_{10}|X_1 X_{10}\sec(X_1)|}{\sqrt{X_2}}) + \sin(\frac{|X_1 X_{10}\sec(X_1)|}{X_{10}}) + \\
& 2\sin(\frac{|\frac{X_1 X_{10}}{\log(\tan(X_1))}|}{X_{10}}) + \sqrt{X_2}\cot(\tan(X_9))\sin(\cos(\frac{\tan(\tan(X_9))}{X_1 X_{10}})) + X_1 X_{10}\cot(\tan(X_9)) \\
& \sin(\sec(X_1)\sin(\frac{\sqrt{X_2}}{X_{10}})) + X_1 X_{10}\sin(X_1)\cot(\tan(X_9)) + X_1 X_{10} + \sin(X_1 \cot(X_9)) + \\
& \left.\left. 2\sin(X_1) + 4\sin(\frac{\sqrt{X_2}}{X_{10}}) + \log(X_{13}) + \sin(\sqrt{X_2})), \tan(X_9))\right)^{\frac{1}{2}}\right)
\end{aligned}
\tag{A11}
$$

$$\begin{aligned}
X_{SPT32} =\ & \log(\min(X_{12}, \sqrt{\min(X_7, X_{11} + \tan(\frac{X_3}{X_7})))}) + \sin(\frac{X_{10}|X_1 X_{10}\sec(X_1)|}{\sqrt{X_2}}) + \\
& \sin(\frac{|X_1 X_{10}\sec(X_1)|}{X_{10}}) + 2\sin(\frac{|\frac{X_1 X_{10}}{\log(\tan(X_1))}|}{X_{10}}) + \sqrt{X_2}\cot(\tan(X_9)) \\
& \sin(\cos(\frac{\tan(\tan(X_9))}{X_1 X_{10}})) + X_1 X_{10}\cot(\tan(X_9))\sin(\sec(X_1)\sin(\frac{\sqrt{X_2}}{X_{10}})) + \\
& X_1 X_{10}\sin(X_1)\cot(\tan(X_9)) + \sin(X_1\cot(X_9)) + \cot(X_1)\tan(X_9) + \\
& 2\sin(X_1) + 3\sin(\frac{\sqrt{X_2}}{X_{10}}) + \sin(\sin(\frac{\sqrt{X_2}}{X_{10}})) + \sin(\sqrt{X_2})
\end{aligned} \tag{A12}$$

*Appendix A.3. Coefficients in Symbolic Expressions for Port Propeller Torque Estimation with Decay State Coefficients*

$$\begin{aligned}
X_{PPTDF11} =\ & \max(X_{10} + X_{12}, X_{15}(\max(X_9(\min(\log(\log(\tan(\log(\tan(X_1)))))), \\
& \min(\tan(X_1), 2X_{10}\log(\log(X_0))|X_{12} + \tan(\log(\tan(\tan(X_1))))|)) + \\
& \log(X_0) + X_{15}(X_{14}(\tan(X_1) + X_{13}X_6) + \tan(X_1) + X_9) + \tan(X_1)) + \\
& \min(\log(\log(\tan(\log(\tan(\tan(X_1)))))), \tan(X_1)) + 2\log(\log(\tan(\log(X_{15} - X_0)))) + \\
& \log(X_9 - X_0) - \sqrt{-\sin(X_0 - X_9)} + 3\log(X_0) + 3\tan(X_1) + \\
& 5\log(\tan(\tan(X_1))) - X_{15}, \log(\tan(X_1)) + 3\tan(\log(X_1)) + 2X_{12} + \\
& \log(\tan(\log(\sqrt{-\sin(X_0 - X_{15})} - X_0)))) + \tan(X_1))
\end{aligned} \tag{A13}$$

$$\begin{aligned}
X_{PPTDF21} =\ & (-X_{14} - \cos(X_{14}) + X_8)\min((X_8 - X_{14})^2\tan(\log(\tan(X_0)(X_8 - X_{14})^3 \\
& \tan(\log((X_8 - 0.999352)(X_8 - X_{14})(X_8 - |X_{15}|))))), X_0) + \\
& \log((X_8 - X_{14})^3\log(X_0(X_8 - X_{14}))\tan(\log(X_1 X_{10}))\tan(\log(X_0(X_8 - X_{14})))) + \\
& \log(X_0 X_8(X_8 - X_{14})^3(\cos(X_0) - X_{14} + X_8)\tan(\log(X_1 X_{10}))\tan(\log(X_0(X_8 - X_{14}))))
\end{aligned} \tag{A14}$$

$$\begin{aligned}
X_{PPTDF31} =\ & \frac{X_{14}}{X_0(-0.181111|\sin(X_1)| - 0.181111|\sin(\sin(X_1))| + X_{13})} + \\
& \frac{X_{13}^4 X_{15}^{15}}{X_9^{13}|\sin(X_1)|^2} + \frac{X_{13}^2 X_{15}}{\sin(X_{10}) + \log(X_8)} + X_{10} - \frac{4.19971 X_9}{X_{13}} + 0.004 X_4
\end{aligned} \tag{A15}$$

*Appendix A.4. Coefficients in Symbolic Expressions for Port Propeller Torque Estimation without Decay State Coefficients*

$$\begin{aligned}
X_{PPT11} =\ & \left(\max(X_{10}, X_{12}) + \left(X_3\cos(X_1(X_9 - 0.002) + \sqrt{X_1})\cos(\sqrt{X_1\cos(X_1 + X_{11})} + \right.\right. \\
& \left.\left. X_1 X_9)\cos\left(\frac{\cos\left(\sqrt{X_{12}(X_1(X_9 - 0.002)X_9 + \sqrt{X_1})}\cos(\frac{\cos(X_1 + X_{11})}{X_{11} + X_9}) + X_{11} + X_9\right)}{\cos(\frac{X_{11}}{X_{11} + X_9}) + X_9}\right)\right)^{\frac{1}{2}}\right)
\end{aligned} \tag{A16}$$

$$\begin{aligned}
X_{PPT21} =\ & (\log(\log(\frac{\log(\log(\min(\frac{X_2}{X_5^2}, \log(\log(\frac{X_0}{X_5}) + \csc(X_0)\log(X_7)))) + \csc(X_0)\log(X_7))}{X_5}) + \\
& \csc(X_0)\log(X_7)))
\end{aligned} \tag{A17}$$

$$X_{PPT31} = \left(\frac{X_{12}^2\sin(\sin(X_0))\log\left(\sqrt{X_{12}}\sin(X_{12})\left(\frac{X_{12}^{5/2}\sin(X_{12})}{X_1} + X_{13}X_6\right)\right)}{X_1^2} + \sqrt{X_6}\right) \tag{A18}$$

$$X_{PPT32} = \tan(\cos(X_{10})) + 2X_{11} - X_{13}X_6 + 2\tan(X_{13}) + \sqrt{X_6} \tag{A19}$$

*Appendix A.5. Coefficients in Symbolic Expressions for Total Propeller Torque Estimation with Decay State Coefficients*

$$
\begin{aligned}
X_{TTDF1} = &\left( \min\left( X_0, \log\left( \sqrt{X_2} \right), X_{12}\sin(\log(\log(X_{12}))) \right) + \log(X_3) \right) \\
&\left( X_{12}\sin\left( \log\left( \min\left( X_0, \log\left( \sqrt{X_2} \right), \log(X_3)\sin(\log(X_0)) \right) \right) \right) \right) \\
&- \sqrt{\log(X_1)}\left( \sqrt{-\sin(X_0 - X_{13})} - \tan(X_{15}) \right) \\
&\tan\left( \min\left( X_{15}, \log\left( \sqrt{\sqrt[4]{\log(X_1)}\sqrt{\log(X_3)} - X_0} \right. \right. \right. \\
&\left. \left. \left. \sqrt{\tan\left( \min\left( X_0, \log\left( \sqrt{X_2} \right) \right) \right)} \right) \right) \right)
\end{aligned} \tag{A20}
$$

*Appendix A.6. Coefficients in Symbolic Expressions for Total Propeller Torque Estimation without Decay State Coefficients*

$$
\begin{aligned}
X_{TT11} = &\left( -\sqrt{X_{10}} - \tan\left( \log(X_{10}) + \tan\left( \sqrt{X_{10}} \right) \right) + \right. \\
&\left. X_{12} - \sqrt{\frac{X_3}{X_8}} - \tan(\tan(X_8)) - X_9 \right)
\end{aligned} \tag{A21}
$$

$$
\begin{aligned}
X_{TT12} = \max &\left( -\left| \left| \left| X_{12} - \tan(\log(X_{10}) + \sqrt{X_{10}}) \right| - \tan(X_8) \right| - \tan(\sqrt{X_{10}}) \right. \right. \\
&\left. - \sqrt{X_{12} - X_9 - \tan(\log(X_{10}) + \tan(\sqrt{X_{10}})) - \sqrt{X_{10}} - \sqrt{\frac{X_3}{X_8}}} \right| \\
&- |X_{12} - 2\tan(\sqrt{X_{10}}) - \tan(X_8) - \tan(X_9 + \sqrt{X_{10}})| + |X_6|, \\
&\left. X_{12} - \sqrt{\frac{X_3}{\tan(X_8)}} \right)
\end{aligned} \tag{A22}
$$

$$
\begin{aligned}
X_{TT21} = \max(&|\max(|\cos(\sqrt{\frac{-|X_5| - X_{12}|X_8| + \frac{\sqrt{X_2} - X_{12}(X_{12}|X_8| + \sqrt{X_2})}{\sqrt{X_2}} - \sqrt{X_2}}{X_{12}|X_8| + \sqrt{X_2}}})|, \cos(X_6))|, \\
&\log(X_{10}) - |\max(\cos(X_6), \cos(\sqrt{\frac{-X_{12}X_8 - |X_5|}{X_{12}}}))| \\
&(\frac{\log(X_{10}) - X_{12}(\log(X_{10})\min(X_{12}, X_3) + \sqrt{X_2})}{\sqrt{X_2}} + X_{12}|X_8|))
\end{aligned} \tag{A23}
$$

$$
\begin{aligned}
X_{TT31} = \min &\left( \log(X_5), \left( \min(X_{12}, -\min(X_1 - 2X_{10} + 2\sqrt{X_{12}}, \sqrt{\cos(\sqrt{X_3})}) + \right. \right. \\
&\left. \left. \sqrt{\min(X_1 + \sqrt{X_1} - X_{10}, \sqrt{X_{12}})} + X_1 - X_{10} \right)^{\frac{1}{2}} + X_1 - X_{10} \right)
\end{aligned} \tag{A24}
$$

$$
\begin{aligned}
X_{TT32} = \Bigg| &\sin\Big(\sin\Big(\sin\Big(\sin\Big(\sin\Big(\sin\Big(\sin\Big(\sin\Big(\sin\Big(\sin\Big(\sin\Big(\sin \\
&\Big(\sin\Big(\sin\Big(\sin\Big(\sin\Big(\sin\Big(\sin\Big(\sin\Big(\sin\Big(\sin\Big(\sin \\
&\Big(\sin\Big(\sin\Big(\sin\Big(\sin\Big(\frac{\cos X_9}{X_{12}}\Big)\Big)\Big)\Big)\Big)\Big)\Big)\Big)\Big)\Big)\Big)\Big)\Big)\Big)\Big)\Big)\Big)\Big)\Big)\Big)\Big)\Big)\Big)\Big)\Big)\Big) \Bigg|
\end{aligned} \tag{A25}
$$

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
