# Peer review of "Use of Genetic Programming for the Estimation of CODLAG Propulsion System Parameters"

_jmse, doi:10.3390/jmse9060612_

Round 1

Reviewer 1 Report

The concept is interesting and the methodology is well presented. The results are clearly given and adequately discussed. Paper can be publishable.

Author Response

Respected reviewer,

we would like to thank you for the review of our manuscript.

Kindest regards,
The authors

Reviewer 2 Report

Most of the comments have been addressed. A couple of minor suggestions to the authors to improve readability.

  1. Try to make Figures 5 and 6 bigger. The distinction between cells is also unclear sometimes, especially with the high number of decimal places used. They data may be presented with columns and row delineation as in a normal table for clarity
  2. Please align the tables such as 1, 12 , 13 etc with the rest of the text

Author Response

Respected Reviewer,

we thank you for your revision of our manuscript. Please find answers to your comments below:

1. Try to make Figures 5 and 6 bigger. The distinction between cells is also unclear sometimes, especially with the high number of decimal places used. They data may be presented with columns and row delineation as in a normal table for clarity

The figures have been made larger for better visibility.

2. Please align the tables such as 1, 12 , 13 etc with the rest of the text

The tables have been aligned throughout the manuscript.

Kindest regards,
the authors

Reviewer 3 Report

The paper deals with the Combined Diesel-Electric and Gas propulsion system outlining the benefit of using genetic programming algorithm to obtain symbolic expressions for estimation of fuel flow, ship speed, starboard propeller torque, port propeller torque, and total propeller torque.

The presentation is extensive and comprehensive. The paper's novelty is the generation of equations that can be applied to the prediction of the most important propulsion system parameters. 

Author Response

Respected reviewer,

thank you for the swift review of our manuscript.

Kindest regards,
the authors.

This manuscript is a resubmission of an earlier submission. The following is a list of the peer review reports and author responses from that submission.

Round 1

Reviewer 1 Report

The paper is generally well written with minor changes recommended for presentation and clarity:

The main comment on content is that the authors can explain further on how the symbolic expressions used in the paper came to be proposed. Were they from trial and error or did they come from a basic expression based on other references etc?

Abstract: Please use the full form of abbreviations CODLAG, MAE, R^2 etc. in the abstract in the 1st instance of usage 

Line 19: Prime movers may be a better term to use as compared to 'propulsion engines' here

Line 114: 'exist' remove the 's'

Lines 169 and 172: Are the 2nd parts of these sentences starting with 'higher levels of one variable...' supposed to be the same for positive and negative values of r?

Equations (5) and (6): Which of these formula will be used to calculate rs in the paper and why?

Figure 4: Could this be labeled as a table? The colors used in the cells do not seem compatible with the rest of the paper. Please consider using lighter shades or color the font only

Table 3, 5th row: Is this supposed to be 'Tree' rather than 'Three'?

Table 4: This table may not be needed, the mathematical symbols appear to be standard shorthand writing commonly used and understood

Table 5: The numbers in the 1st column makes the presentation cluttered. What GP parameters do they represent? Please consider presenting them in another manner

Line 328: Was it intended to be 'tree depth'?

Line 432: 'It can be noted that...'

Line 437: '...these two variables represent ambient temperature...'

Line 516: 'the' is not needed here

Author Response

We thank the first reviewer for their review of our manuscript. We have included the requested information - both in the manuscript and in the answers to posed comments below, with corrections being made according

The paper is generally well written with minor changes recommended for presentation and clarity:

The main comment on content is that the authors can explain further on how the symbolic expressions used in the paper came to be proposed. Were they from trial and error or did they come from a basic expression based on other references etc?

The equations in the paper are developed using the afformentioned GP algorithm, which is a stochastic process, generating them randomly through the evolutionary process. The following explanation has been added into the paper to explain this more clearly:

“The equations are not based on previous knowledge or derived from other findings - but generated purely through the evolutionary process of GP described in the Methodology, which attempts to, in a heuristic manner, develop equations that provide a high fitness value for the used dataset.”

Abstract: Please use the full form of abbreviations CODLAG, MAE, R^2 etc. in the abstract in the 1st instance of usage 

The full forms were added, with the abbreviations in the brackets.

Line 19: Prime movers may be a better term to use as compared to 'propulsion engines' here

The suggested terminology has been used.

Line 114: 'exist' remove the 's'

The 's' has been removed.

Lines 169 and 172: Are the 2nd parts of these sentences starting with 'higher levels of one variable...' supposed to be the same for positive and negative values of r?

To clarify, the “higher levels” have been changed to “higher absolute levels”.

Equations (5) and (6): Which of these formula will be used to calculate rs in the paper and why?

The following explanation was provided:

“To avoid the step of determining the ranks of the variables, the Equation \ref{eq:spearman} has been used for the calculation of Spearman's correlation coefficients in this paper”.

Figure 4: Could this be labeled as a table? The colors used in the cells do not seem compatible with the rest of the paper. Please consider using lighter shades or color the font only

    The images have been changed to increase the visibility.

Table 3, 5th row: Is this supposed to be 'Tree' rather than 'Three'?

Yes it was, the text was modified as instructed.

Table 4: This table may not be needed, the mathematical symbols appear to be standard shorthand writing commonly used and understood

The table in question has been removed.

Table 5: The numbers in the 1st column makes the presentation cluttered. What GP parameters do they represent? Please consider presenting them in another manner

The names of individual hyperparameters have been added into the table header, for all tables presenting results with hyperparameters.

Line 328: Was it intended to be 'tree depth'?

Yes, thank you - the text was modified as instructed.

Line 432: 'It can be noted that...'

    The text was modified as instructed.

Line 437: '...these two variables represent ambient temperature...'

The text was modified as instructed.

Line 516: 'the' is not needed here

The text was modified as instructed.

---

Kindest regards,

the authors.

Reviewer 2 Report

Thank you for inviting me to review the manuscript below:

Journal: JMSE (ISSN 2077-1312)

Manuscript ID: jmse-1223724

Type: Article

Number of Pages: 30

Title: Fuel flow, Ship speed, Starboard, Port and Total Propeller Torque Estimation of CODLAG Propulsion System Using Genetic Programming Algorithm

The concept is interesting, the methodology is well presented but the paper needs some major revisions:

  1. The title should be more interesting and attractive.
  2. The novelty, originality shall be further justified that the manuscript contains sufficient contributions to the new body of knowledge. The knowledge gap needs to be clearly addressed in the Introduction. At the end of the introduction Add an article structure.
  3. All already known equations must be cited. Eq. (7) should be cited with the article https://doi-org.ezproxy.lib.ukm.si/10.1016/j.energy.2014.11.074
  1. In the Description section please add the thermodynamic chart of the CODLAG system.
  2. The results and discussion should be discussed more extensively by increasing the number of literature. This article should be cited in the net benefit section. https://doi.org/10.1016/j.ijheatmasstransfer.2020.119897
  3. In the text there are errors in English, need to be carefully read and corrected.

Author Response

The authors would like to thank the second reviewer for their review of our manuscript. We have attempted to answer the posed comments to the best of our abilities. We sincerely hope that the changes introduced to the manuscript will make it worthy of publication. Please find the answers to the comments below.

---

#1 The title should be more interesting and attractive.

The title has been changed and is now: “Utilization of Genetic Programming for the Estimation of CODLAG Propulsion System Parameters”.

#2 The novelty, originality shall be further justified that the manuscript contains sufficient contributions to the new body of knowledge. The knowledge gap needs to be clearly addressed in the Introduction. At the end of the introduction Add an article structure.

The novelty has been addressed with the following paragraph:

“Novelty of the presented research is multi-fold, with the first novel element being the application of GP for the determination of symbolic expressions used to regress the CODLAG system parameters. This is followed by the correlation analysis applied as preprocessing to the dataset to allow faster convergence to higher quality solutions for the GP. Finally, the testing of the influence of the decay factor inclusion on the output of the models is tested and determined.”

Article structure is addressed with the following paragraph in introduction:

“First, the researchers will present the used dataset, with methods applied to the analysis of it. Then, the short description of GP algorithm is provided, along with the used hyperparameters and evaluation metrics. The results are presented and discussed following that, providing information on the correlation coefficients of the parameters in the dataset, metrics achieved with the trained models along the used hyperparameters and regressed equations. Drawn conclusions, addressing the posed research questions, are given in the end.”

#3 All already known equations must be cited. Eq. (7) should be cited with the article https://doi-org.ezproxy.lib.ukm.si/10.1016/j.energy.2014.11.074

The appropriate citations, including the suggested one, were added to the equations 1, 3, 4, 5, 6, 7, and 8.

#4 In the Description section please add the thermodynamic chart of the CODLAG system.

As the analyzed CODLAG propulsion system consists of two independent diesel engines and the gas turbine (along with gearboxes and clutches) we don't have an idea how to present a thermodynamic chart of the whole CODLAG propulsion system. Also, in the literature we did not find any example of such a presentation.

The only possibility was to present a chart for the whole CODLAG system with various energy flows (mechanical energy and electrical energy), but such a presentation will not show much more in comparison to already presented elements in Figure 1. Therefore, we believe that the Reviewer will be satisfied with the presentation of the gas turbine thermodynamic process from which can be seen all the losses which occur in the gas turbine process. Also, the analysis takes into account various operating regimes, so the presentation of fluid flow streams also will not result with appropriate presentation (such presentation will show only one system load).

Moreover, the dominant and most important operating parameters related to CODLAG propulsion system are actually gas turbine operating parameters, as presented in Table 1. Using these operating parameters, along with the application of Genetic Programming allows  tracking the dominant operating parameters of the whole CODLAG propulsion system. Mentioned operating parameters are the elements from the publicly available dataset – any details related to diesel engines were not known, therefore any presentation of diesel engine operation will be highly questionable and possibly wrong. The authors have decided to avoid such questionable presentations – with an aim to make the whole paper clear and easily understandable.

#5 The results and discussion should be discussed more extensively by increasing the number of literature. This article should be cited in the net benefit section. https://doi.org/10.1016/j.ijheatmasstransfer.2020.119897

The following paragraph was added inside the section in question, with added literature - including the suggested paper:

“In comparison to the previous work in the field it can be seen that GP implementation  in this paper achieves comparable results to other works using it. The same can be said for other researches with similar goals, such as in which the used methods achieve results which are comparable to the ones achieved by GP. Comparison to the performance of the existing work in AI-based CODLAG system modelling, which utilized other ML algorithms it is seen that results achieved by  GP are comparable or better, with the benefit of clearer models. The clearer models in question make it possible to see which of the inputs (such as decay coefficients) ended up not being included in the best performing models signifying their low influence in the final model.”

#6In the text there are errors in English, need to be carefully read and corrected.

The manuscript has been thoroughly checked with corrections being applied where necessary.

---

Kindest regards,

The authors

Reviewer 3 Report

This paper needs to make a significant changes. 

1. In the keyword section, correlation analysis is not a good keyword

2. The introduction is too long. I cannot catch up the key context, the motivation of reading this paper, the research objectives. Please narrow down. 

3. Please provide the paper structure at the end of the introduction section. 

4. I cannot see the clear literature reviews in this paper. Please provide it. 

5. In the discussion, there is no literature review support. The authors only based on the data analysis result to present the findings. 

6. In the conclusion, it only summarize the key findings. There is a serious lack of academic and managerial implications. The weakness and future of research direction are also overlooked. 

7. To conclude, I think that this paper looks like a technical report rather than an academic paper. 

Author Response

The authors would like to thank the third reviewer for their review of our paper. We have made changes according to the received comments, answers to which follow below. We sincerely hope that the third reviewer will consider our manuscript worthy of publication after the revisions.

---

  1. In the keyword section, correlation analysis is not a good keyword

    “Correlation Analysis” has been replaced with “Data-driven Modelling”

  1. The introduction is too long. I cannot catch up the key context, the motivation of reading this paper, the research objectives. Please narrow down. 

The introduction has been shortened and rewritten to better emphasize the context and motivation of the papers.

  1. Please provide the paper structure at the end of the introduction section. 

Article structure is addressed with the following paragraph in introduction:

“First, the researchers will present the used dataset, with methods applied to the analysis of it. Then, the short description of GP algorithm is provided, along with the used hyperparameters and evaluation metrics. The results are presented and discussed following that, providing information on the correlation coefficients of the parameters in the dataset, metrics achieved with the trained models along the used hyperparameters and regressed equations. Drawn conclusions, addressing the posed research questions, are given in the end.”

  1. I cannot see the clear literature reviews in this paper. Please provide it. 

The literature review section in the Introduction has been expanded through the addition of new papers.

  1. In the discussion, there is no literature review support. The authors only based on the data analysis result to present the findings. 

The following paragraph was added in the manuscript to address the reviewer question:

“In comparison to the previous work in the field it can be seen that GP implementation  in this paper achieves comparable results to other works using it. The same can be said for other researches with similar goals, such as in which the used methods achieve results which are comparable to the ones achieved by GP. Comparison to the performance of the existing work in AI-based CODLAG system modelling, which utilized other ML algorithms it is seen that results achieved by  GP are comparable or better, with the benefit of clearer models. The clearer models in question make it possible to see which of the inputs (such as decay coefficients) ended up not being included in the best performing models signifying their low influence in the final model.”

  1. In the conclusion, it only summarize the key findings. There is a serious lack of academic and managerial implications. The weakness and future of research direction are also overlooked. 

The following paragraph was added in order to better explain the points the reviewer mentions in the conclusion:

“The findings of the paper demonstrate the ability of the application of GP for the regression of the CODLAG system parameters. Academical applications are the possibility for the utilization of the determined equations for a precise determination of the regressed system parameters. Such an approach can greatly decrease the time necessary for the modeling of the system at various operating points.  The use of GP as opposed to different AI-based modeling techniques is the shape of the generated models, which are mathematical equations, that can be easily and more simply implemented within existing or newly developed systems as they are not limited to an individual programming language or a specific library as is commonly the case. While only the CODLAG system in particular is modeled, the approach may be applied to different propulsion systems for which the data can be collected in future work.”

  1. To conclude, I think that this paper looks like a technical report rather than an academic paper.

With all the respect to the Reviewer, but the authors cannot agree with this statement. In marine propulsion systems – using a Genetic Programming (GP) is a novel element – instead or our own, in the literature cannot be found much (or any) research which presents implementation of GP in such systems. The data were not only collected and analyzed (as in technical reports) – demanding computing was performed to finally get direct equations which connects some of the operating parameters from the system with the most important outputs. Such equations can allow much easier tracking the performance of the system and can lead to measurement equipment reducing. Therefore, a practical implementation of such equations onboard a ship can be very interesting to various ship owners.

In our opinion, if the research is performed by using a novel approach, if it is rare and if it requires extensive computing (along with the interesting applicability options), that is true academic research. We have attempted to make this clearer with the changes to the conclusion, introduction and conclusion of the submitted manuscript.

---

Kindest regards,

The authors

.

Round 2

Reviewer 2 Report

Thank you for inviting me to review the manuscript below:

Journal: JMSE (ISSN 2077-1312)

Manuscript ID jmse-1223724

Title: Fuel flow, Ship speed, Starboard, Port and Total Propeller Torque Estimation of CODLAG Propulsion System Using Genetic Programming Algorithm

This paper focuses on the fuel flow, ship speed, starboard, port and total propeller torque estimation of CODLAG propulsion system using genetic programming algorithm. The concept is interesting and the methodology is well presented. The results are clearly given and adequately discussed. The Paper may be accepted after minor revision.

  1. A grammatical error in Table 1., please replace the pressure with temperature “Turbo compressor outlet air pressure (T2)”.
  1. In Figure 2. please add the thermodynamic points according to the marked points in Figure 3.

Reviewer 3 Report

The paper still not meet the academic journal standard. It is far from my expectation in the revision round. 

  1. The introduction is still long. I have seen that the authors have improved it. But, as an academic paper, it should have a clear literature review section. 
  2. In general, there is no a significant academic knowledge can be found. Thus, I need to consider reject the paper.